# Genetic architecture of heart failure with preserved versus reduced ejection fraction

Jacob Joseph [1,2,3,13] ✉, Chang Liu [4], Qin Hui [4,5], Krishna Aragam [1,6,7], Zeyuan Wang[4,5], Brian Charest[1], Jennifer E. Huffman [1], Jacob M. Keaton[8,9], Todd L. Edwards [10], Serkalem Demissie[1,11], Luc Djousse[1,2], Juan P. Casas[1,2], J. Michael Gaziano[1,2], Kelly Cho[1,2], Peter W. F. Wilson[5,12], Lawrence S. Phillips[5,12], VA Million Veteran Program*, Christopher J. O'Donnell [1,2] & Yan V. Sun [4,5,13] ✉

Pharmacologic clinical trials for heart failure with preserved ejection fraction have been largely unsuccessful as compared to those for heart failure with reduced ejection fraction. Whether differences in the genetic underpinnings of these major heart failure subtypes may provide insights into the disparate outcomes of clinical trials remains unknown. We utilize a large, uniformly phenotyped, single cohort of heart failure sub-classified into heart failure with reduced and with preserved ejection fractions based on current clinical definitions, to conduct detailed genetic analyses of the two heart failure sub-types. We find different genetic architectures and distinct genetic association profiles between heart failure with reduced and with preserved ejection fraction suggesting differences in underlying pathobiology. The modest genetic discovery for heart failure with preserved ejection fraction (one locus) compared to heart failure with reduced ejection fraction (13 loci) despite comparable sample sizes indicates that clinically defined heart failure with preserved ejection fraction likely represents the amalgamation of several, distinct pathobiological entities. Development of consensus sub-phenotyping of heart failure with preserved ejection fraction is paramount to better dissect the underlying genetic signals and contributors to this highly prevalent condition.

Heart failure (HF) affects ~64 million people worldwide and 6.2 million adults in the United States[1,2]. While major advances in therapy have reduced the morbidity and mortality due to heart failure with reduced ejection fraction (HFrEF), there is significant residual risk of adverse outcomes[3]. Therapeutic options are limited for heart failure with preserved ejection fraction (HFpEF), which accounts for approximately half of all cases of HF, with large-scale clinical trials largely failing to demonstrate conclusive benefits[4,5]. Agents that have reduced the progression of myocardial remodeling and reduced adverse outcomes in HFrEF have not demonstrated comparable benefit in HFpEF.

[1]Massachusetts Veterans Epidemiology Research and Information Center, VA Boston Healthcare System, Boston, MA, USA. [2]Department of Medicine, Brigham and Women's Hospital, Harvard Medical School, Boston, MA, USA. [3]Cardiology Section (111A), VA Providence Healthcare System, 830 Chalkstone Avenue, Providence, RI 02908, USA. [4]Emory University Rollins School of Public Health, Atlanta, GA, USA. [5]Atlanta VA Health Care System, Decatur, GA, USA. [6]Massachusetts General Hospital, Boston, MA, USA. [7]Broad Institute of Harvard and MIT, Cambridge, MA, USA. [8]Center for Precision Health Research, National Human Genome Research Institute, National Institutes of Health, Bethesda, MD, USA. [9]Division of Epidemiology, Department of Medicine, Vanderbilt University Medical Center, Nashville, TN, USA. [10]Division of Epidemiology, Department of Medicine, Vanderbilt Genetics Institute, Vanderbilt University Medical Center, Nashville, TN, USA. [11]Boston University School of Medicine, Boston, MA, USA. [12]Emory University School of Medicine, Atlanta, GA, USA. [13]These authors jointly supervised this work: Jacob Joseph, Yan V. Sun. *A list of authors and their affiliations appears at the end of the paper. ✉e-mail: jacob.joseph@va.gov; yan.v.sun@emory.edu

Genomic analyses of large cohorts represent promising approaches to better understand the pathobiology of HFrEF and HFpEF[6,7]. A recent GWAS meta-analysis of multiple cohorts of European ancestry has identified several genomic loci associated with unclassified HF, although similar genomic analyses focused on HFrEF and HFpEF are lacking[8]. The Million Veteran Program (MVP) is a large biobank linked to extensive national Veterans Affairs (VA) electronic health record (EHR) databases. Using algorithms developed to curate HFrEF and HFpEF phenotypes in the national VA databases based on current consensus definitions[9], we extensively explored the genetic architecture of each HF subtype in a single large cohort in the MVP. In addition to demonstrating the disparate genetic underpinnings of HFrEF and HFpEF, our results highlight the marked heterogeneity of the HFpEF phenotype, and the urgent need to develop consensus approaches to subphenotype HFpEF to enable pathophysiological and therapeutic discovery.

## Results

The primary study population for the GWAS consisted of 258,943 controls, and cases of unclassified HF ($n = 43,344$), HFpEF ($n = 19,589$), and HFrEF ($n = 19,495$) from the MVP cohort, and 8227 HF cases and 379,788 controls from the UK Biobank cohort, all of European genetic ancestry. The genome-wide significant (GWS) associations of unclassified HF, HFrEF and HFpEF were then examined in the MVP non-Hispanic African Americans and a recent HF GWAS in Europeans from the HERMES consortium (Figs. 1 and 2). The MVP control and HF cohorts were predominantly male. In both MVP and UK Biobank, the HF cohorts tended to be older with a higher prevalence of cardiometabolic risk factors and comorbidities than the control populations (Table 1 and Supplementary Data 1 and 2).

### GWAS of unclassified HF

In unclassified HF, the meta-analysis of MVP and UK Biobank GWAS results (Supplementary Figs. 1 and 2) identified 20 genome-wide significant (GWS) loci including 10 novel loci (Table 2 and Supplementary Data 3 and 4). The regional association plots of each GWS locus are shown in Supplementary Fig. 3A–T. We replicated all 12 GWS independent SNPs associated with HF from a recent HF GWAS publication[8], (Bonferroni-corrected p-value <0.05; Supplementary Data 5).

### GWAS of HFrEF and HFpEF

We conducted GWAS in cohorts of HFrEF and HFpEF curated based on the current definitions. First, we compared the output of GWAS for the more and less restrictive HFpEF definitions and observed high, overall genetic correlation ($r = 0.981$, $p < 2 \times 10^{-16}$) between these phenotypes, including among the top 110 HFpEF-associated SNPs ($r = 0.995$, $p < 2 \times 10^{-16}$; Supplementary Fig. 4). We therefore used the less restrictive (and better-powered) HFpEF definition as the primary HFpEF phenotype for all subsequent analyses.

In the GWAS among the MVP participants of European ancestry, we identified 13 GWS loci associated with HFrEF and one GWS locus (FTO) associated with HFpEF (Fig. 3; Table 3; Supplementary Fig. 5A, B). The regional association plots of each GWS locus are shown in Supplementary Fig. 6A–N. Two lead SNPs in the FTO locus for HFrEF (rs7188250) and HFpEF (rs11642015) were in linkage disequilibrium ($r^2 = 0.873$). Among these thirteen loci associated with HF subtypes, seven loci (NFIA, E2F6, MITF, PHACTR1, METTL7A, PNMT, and BPTF) have not been reported in previous HF-related GWAS, of which four loci (NFIA, MITF, PHACTR1, and METTL7A) were GWS only in GWAS of HFrEF cases. A scatterplot illustrating the comparison between the effect sizes of the GWS loci for HFrEF and HFpEF is shown in Supplementary Fig. 7, with effect sizes with standard errors for HFrEF and HFpEF on X- and Y-axis, respectively.

Among 13 HFrEF-associated loci, nine loci had different associations with HFrEF and HFpEF (p-value < 0.0038, corrected for 13 tests, Table 3).

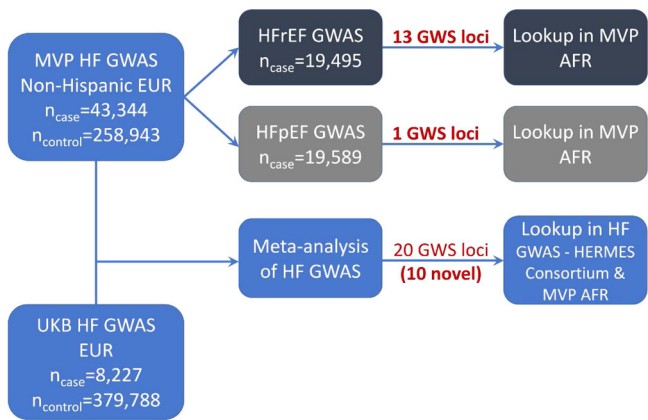

**Fig. 1 | Study schema.** Schematic diagram detailing datasets and analyses used in the study.

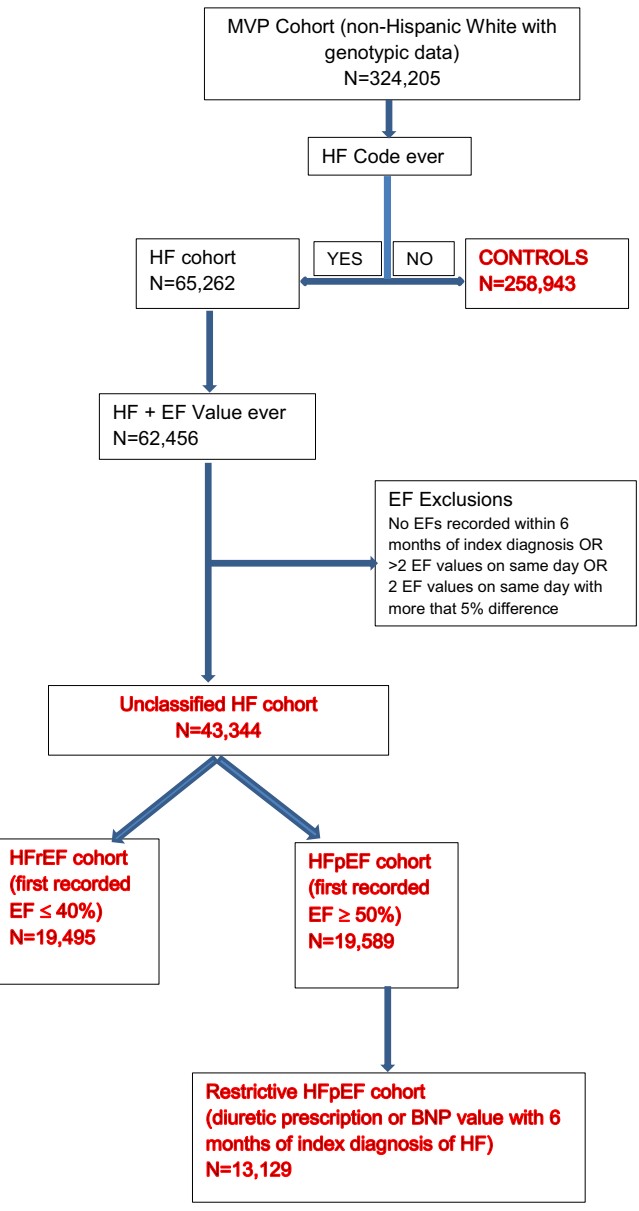

**Fig. 2 | Algorithm for phenotyping of cohorts for genetic analyses.** Consort diagram describes the methodology utilized to accurately phenotype the case cohorts (unclassified HF, HFrEF, HFpEF, restrictive case definition of HFpEF) and controls included in the study from the Million Veteran Program.

**Table 1 | Characteristics of HF patients and non-HF controls in the MVP participants of European Ancestry**

| Group | Control (N = 258,943) | HFpEF (N = 19,589) | HFrEF (N = 19,495) | Unclassified HF (N = 43,344) |
|---|---|---|---|---|
| Age (years), mean±SD | 62.74 ± 13.76 | 69.88 ± 9.77 | 69.29 ± 9.74 | 69.61 ± 9.74 |
| Male (%) | 92.14 | 95.74 | 97.85 | 96.92 |
| Body mass index (kg/m$^2$), mean ± | 29.20 ± 5.53 | 31.95 ± 6.98 | 30.20 ± 6.38 | 31.08 ± 6.73 |
| Underweight (<18.5) % | 0.56 | 0.47 | 0.59 | 0.52 |
| Normal (18.5–24.9) % | 20.25 | 13.43 | 18.79 | 16.05 |
| Overweight (25.0–29.9) % | 40.66 | 29.71 | 35.09 | 32.44 |
| Obese (30.0–34.9) % | 24.70 | 27.08 | 25.62 | 26.37 |
| Morbidly obese (≥35.0) % | 13.84 | 29.31 | 19.91 | 24.62 |
| LVEF, mean ± SD | NA | 56.97 ± 5.65 | 29.33 ± 9.36 | 43.36 ± 15.05 |
| Atrial fibrillation (%) | 6.33 | 30.80 | 37.83 | 34.44 |
| Coronary artery disease (%) | 22.47 | 63.87 | 74.63 | 69.72 |
| Chronic kidney disease (%) | 9.54 | 37.21 | 35.75 | 36.43 |
| Diabetes (%) | 20.61 | 48.54 | 45.06 | 46.76 |
| Hyperlipidemia (%) | 66.9 | 87.75 | 88.20 | 88.04 |
| Hypertension (%) | 62.97 | 93.22 | 91.69 | 92.51 |
| Peripheral vascular disease (%) | 15.18 | 42.47 | 42.27 | 42.47 |
| Stroke/TIA (%) | 8.26 | 25.29 | 24.33 | 24.93 |

*HFpEF* heart failure with preserved ejection fraction, *HFrEF* heart failure with reduced ejection fraction, *HF* heart failure, *SD* standard deviation, *LVEF* left ventricular ejection fraction, *TIA* transient ischemic attack.

For example, the risk allele of the *BAG3* missense variant (rs2234962) was associated with higher risk for HFrEF (OR 1.12, 95% CI 1.09–1.15, p-value 9.02 × 10$^{-18}$), but was associated with lower risk for HFpEF (OR 0.97, 95% CI 0.94–0.99, p-value 6.42 × 10$^{-3}$). Only four loci, including *LPA*, *FTO*, *PNMT*, and *BPTF*, were not differentially associated with HF subtypes.

We observed moderate genomic inflation ($\lambda$) for unclassified HF ($\lambda = 1.263$), HFrEF ($\lambda = 1.152$), and HFpEF ($\lambda = 1.118$), on par with GWAS of phenotypes with similarly large sample sizes. The LDSC intercepts were 1.044 (SE 0.010), 1.013 (SE 0.008), and 1.028 (SE 0.008) for unclassified HF, HFrEF, and HFpEF, respectively, indicating that most of the inflation was due to polygenicity of HF and subtypes.

### Replication in MVP African Americans and other HF GWAS

Among MVP African Americans, all but two of the SNPs identified in the GWAS of unclassified HF in the European ancestry had genetic associations with unclassified HF in the same direction, and two (rs3176326-CDKN1A and rs12150603-PNMT) were significant after Bonferroni correction (Supplementary Data 4); four (rs4717903-GTF2I, rs12933292-NFAT5, rs1002135-SMG6, and rs1999323-MAP3K7CL) were replicated in the recent HF GWAS[8] after Bonferroni correction.

Among 13 GWS loci associated with HFrEF, 11 had genetic effects in the same direction in the MVP African American cohort (Supplementary Data 6), including three (rs1763610-HSPB7, rs4151702-CDKN1A, and rs2234962-BAG3) which were test-wise significant after Bonferroni correction. Interestingly, the sentinel SNP of the *FTO* locus was significantly associated with HFpEF (rs11642015, OR 1.10, 95% CI 1.03–1.17, p-value 6.30 × 10$^{-3}$), but not associated with HFrEF (rs7188250, OR 1.06, 95% CI 0.99–1.12, p-value 0.11).

### Genetic associations with HFrEF and HFpEF in candidate genes and loci

Out of 12 GWS loci reported in the recent HERMES study of unclassified HF, all were associated with HFrEF, but only four were significantly associated with HFpEF including the *FTO* locus (Supplementary Data 5). Other loci replicated in HFrEF were *ZBTB17/HSPB7* locus (closest gene of *SRARP* discovered in our study) and *HCG22* locus[10] (OR 1.05, CI 1.03–1.08, P = 7.83 × 10$^{-5}$). We did not replicate previously reported associations of *FRMD4B* or *USP3* region with HF[6,11]. Among 17 autosomal genes related to cardiomyopathy[12,13],

we found significant associations in HFrEF with *TMEM43, BAG3, MYBPC3, TTN*, and in *HFpEF* with *DSG2* and *PRKAG2* (Supplementary Data 7, Supplementary Fig. 8).

### Associations of HFrEF- and HFpEF loci with cardiovascular risk factors

As shown in Fig. 4 and Supplementary Data 8, several of the 13 loci associated with HFrEF and HFpEF also demonstrated genetic associations with risk factors as previously reported (*PHACTR1*, *LPA*, and *CDKN2B-AS* with CAD; *CDKN1A* with AF); and *FTO* with BMI, T2D, and HDL cholesterol. Although most loci were associated with multiple risk factors, the *BAG3* locus was only associated with blood pressure traits, and the *MITF* and *METTL7A* loci were associated with eGFR. Three novel loci, *SRARP*, *NFIA*, and *E2F6*, were not significantly associated with any tested HF risk factors. Genome-wide significant loci for unclassified HF and subtypes associated with ~2400 traits tested in the UK Biobank (searched in PheWeb browser, https://pheweb.org/) with p < 1 × 10$^{-6}$ are listed in the Supplementary Data 9.

### Genetic correlation between HFrEF and HFpEF and heritability

Using LDSC and the MVP GWAS summary statistics, we estimated the heritability ($h^2$) of unclassified HF, HFpEF and HFrEF as 3.7% (SE 0.3%), 1.9% (SE 0.2%), and 3.1% (0.3%), respectively. Heritability of HFpEF was substantially lower than that of unclassified HF and HFrEF. We also identified a modest positive genetic correlation between HFrEF and HFpEF (0.57 ± 0.07). The LDSC ratios for unclassified HF, HFrEF, and HFpEF are 0.1381 (SE of 0.0295), 0.0723 (SE of 0.0456), and 0.2184 (SE of 0.0592), respectively.

We estimated the SNP-based heritability using GREML-LDMS-I in MVP non-Hispanic Whites. Assuming a prevalence of HFrEF and HFpEF of 2.5%, 5%, and 7.0% in the population, we derived similar heritability on the liability scale between HFrEF (0.25, 0.31, 0.34, respectively) and HFpEF (0.22, 0.26, 0.29, respectively) (Supplementary Fig. 9).

### Mendelian randomization association analysis of HF risk factors

We present the MR association results from the inverse-variance-weighted method (Fig. 5) since the assumption of zero-intercept was not violated in the Egger regression (Supplementary Data 10 shows results of all 3 MR methods). In primary MR analyses (inverse-variance-weighted estimates), CAD had a stronger causal association with

**Table 2 | Sentinel SNPs significantly associated with heart failure**

| rsID | Position | Closest gene (* denotes novel association) | Genomic region | Risk allele/Ref. allele | Risk allele frequency | META HF GWAS | | MVP HF GWAS | |
|---|---|---|---|---|---|---|---|---|---|
| | | | | | | OR (95% CI) | p-value | OR (95% CI) | p-value |
| rs371236917 | 1:16310737 | SRARP/HSPB7/ZBTB17 | Flanking | C/CT | 0.70 | 1.06 (1.05, 1.08) | $4.97 \times 10^{-15}$ | 1.06 (1.04, 1.08) | $1.50 \times 10^{-12}$ |
| rs1277930 | 1:109822143 | CELSR2 | Flanking | A/G | 0.77 | 1.05 (1.04, 1.07) | $1.10 \times 10^{-10}$ | 1.05 (1.03, 1.07) | $7.20 \times 10^{-8}$ |
| rs7595697 | 2:11568158 | E2F6* | Flanking | T/C | 0.37 | 1.04 (1.02, 1.05) | $4.98 \times 10^{-8}$ | 1.04 (1.02, 1.05) | $1.05 \times 10^{-6}$ |
| rs6795366 | 3:44005735 | ABHD5* | Intergenic | C/T | 0.74 | 1.05 (1.03, 1.06) | $1.95 \times 10^{-8}$ | 1.04 (1.02, 1.06) | $5.36 \times 10^{-6}$ |
| rs2634073 | 4:111665783 | PITX2 | Intergenic | T/C | 0.20 | 1.08 (1.06, 1.10) | $7.42 \times 10^{-19}$ | 1.07 (1.05, 1.09) | $1.63 \times 10^{-11}$ |
| rs3176326 | 6:36647289 | CDKN1A | Intron | G/A | 0.80 | 1.08 (1.06, 1.10) | $1.08 \times 10^{-18}$ | 1.08 (1.06, 1.10) | $1.00 \times 10^{-15}$ |
| rs10455872 | 6:161010118 | LPA | Intron | G/A | 0.07 | 1.11 (1.08, 1.14) | $9.34 \times 10^{-17}$ | 1.11 (1.08, 1.14) | $7.73 \times 10^{-13}$ |
| rs4717903 | 7:74068167 | GTF2I* | Flanking | C/T | 0.25 | 1.04 (1.03, 1.06) | $3.55 \times 10^{-8}$ | 1.04 (1.02, 1.06) | $1.33 \times 10^{-5}$ |
| rs4977575 | 9:22124744 | CDKN2B-AS | Intergenic | G/C | 0.49 | 1.07 (1.06, 1.09) | $6.92 \times 10^{-23}$ | 1.06 (1.05, 1.08) | $3.87 \times 10^{-16}$ |
| rs579459 | 9:136154168 | ABO | Flanking | C/T | 0.22 | 1.05 (1.03, 1.06) | $1.26 \times 10^{-8}$ | 1.04 (1.02, 1.06) | $3.43 \times 10^{-6}$ |
| rs59693993 | 10:75583034 | CAMK2G | Intron | C/T | 0.86 | 1.06 (1.04, 1.08) | $3.08 \times 10^{-8}$ | 1.05 (1.03, 1.08) | $1.79 \times 10^{-6}$ |
| rs61869036 | 10:121422836 | BAG3 | Intron | G/C | 0.79 | 1.06 (1.04, 1.08) | $1.76 \times 10^{-11}$ | 1.04 (1.03, 1.06) | $3.13 \times 10^{-6}$ |
| rs12149832 | 16:53842908 | FTO | Intron | A/G | 0.41 | 1.07 (1.05, 1.08) | $3.40 \times 10^{-21}$ | 1.07 (1.06, 1.09) | $9.05 \times 10^{-20}$ |
| rs12933292 | 16:69566309 | NFAT5* | Intergenic | C/G | 0.59 | 1.04 (1.03, 1.06) | $5.25 \times 10^{-9}$ | 1.04 (1.03, 1.06) | $2.75 \times 10^{-7}$ |
| rs1002135 | 17:2097583 | SMG6* | Intron | G/T | 0.38 | 1.04 (1.03, 1.06) | $6.33 \times 10^{-9}$ | 1.04 (1.02, 1.06) | $5.89 \times 10^{-7}$ |
| rs12150603 | 17:37834715 | PNMT/PGAP3* | Intron | G/A | 0.35 | 1.04 (1.03, 1.06) | $5.21 \times 10^{-9}$ | 1.05 (1.03, 1.06) | $5.78 \times 10^{-8}$ |
| rs150947345 | 17:57486425 | YPEL2* | Flanking | A/T | 0.02 | 1.16 (1.10, 1.22) | $1.67 \times 10^{-8}$ | 1.16 (1.09, 1.23) | $1.62 \times 10^{-6}$ |
| rs34432450 | 17:65880259 | BPTF* | Intron | C/T | 0.21 | 1.06 (1.04, 1.08) | $1.93 \times 10^{-12}$ | 1.06 (1.04, 1.08) | $3.30 \times 10^{-10}$ |
| rs79329549 | 18:36560942 | 18q12.2* | Intergenic | T/G | 0.91 | 1.07 (1.05, 1.10) | $4.60 \times 10^{-9}$ | 1.08 (1.05, 1.11) | $5.24 \times 10^{-8}$ |
| rs1999323 | 21:30534128 | MAP3K7CL* | Intron | T/C | 0.15 | 1.07 (1.05, 1.09) | $5.26 \times 10^{-11}$ | 1.05 (1.03, 1.07) | $4.04 \times 10^{-6}$ |

Chromosomal position is based on GRCh37/hg19 reference. The sentinel SNPs were mapped to the closest refseq genes based on chromosomal base-pair position. All genetic associations were aligned to effects of the risk alleles (i.e., increased risk for unclassified HF).

Ref reference, OR odds ratio, CI confidence interval, GWAS genome-wide association study, MVP Million Veteran Program cohort ($n_{cases}$ = 43,344), META meta-analysis of MVP and UK Biobank cohorts.

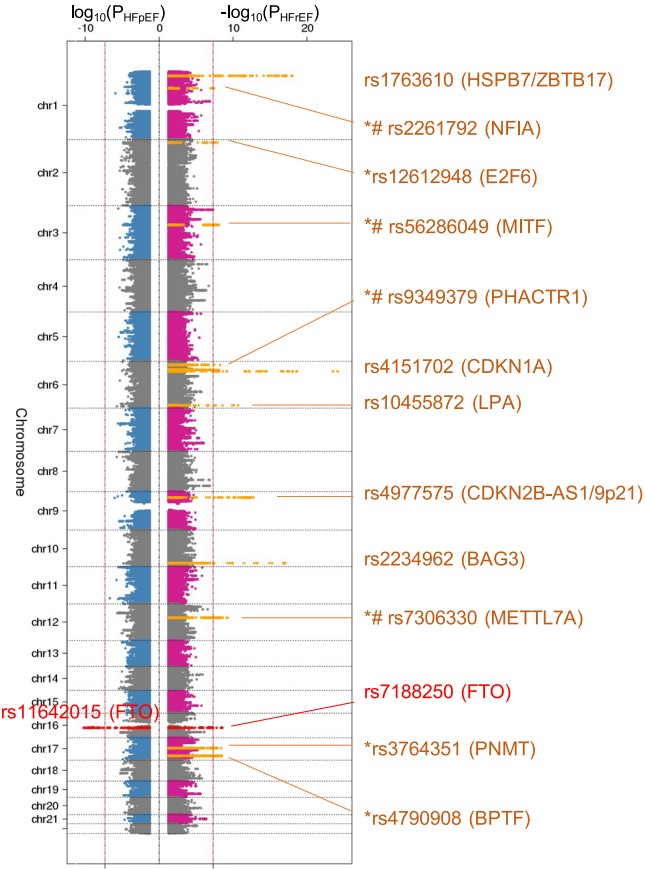

**Fig. 3 | Genome-wide associations of HFrEF and HFpEF.** Genome-wide significant loci association studies of HFpEF and HFrEF among non-Hispanic White veterans. Sentinel SNPs and the nearest mapped genes are shown. *Y*-axis shows chromosomal position. Sentinel SNPs and their nearest genes are shown. All tests were two-sided without adjustment for multiple comparisons. *: novel HF locus; #: unique locus in the HFrEF GWAS but not in the HF meta-analysis; dashed vertical line indicates genome-wide significance threshold ($P = 5 \times 10^{-8}$).

HFrEF, and all lipid parameters as well as T2D and DBP had a significant causal association only with HFrEF. While AF, BMI, and SBP demonstrated similar causal associations with both HF subtypes, PP was significantly associated with HFpEF only. Similar results were observed from the median weighted method (Supplementary Data 10). Sensitivity analysis using Egger regression showed consistent effect estimates but larger confidence intervals (Supplementary Data 10).

#### Conditional analysis and credible set analysis
We identified a secondary SNP in two loci on chromosome 4 and 6 after conditional analysis on the sentinel SNP from unclassified HF GWAS (Supplementary Data 11). However, there was no evidence of secondary independent variants at any GWS loci of HF subtypes in conditional analyses.

We performed a credible set analysis of all GWS loci for unclassified HF, HFrEF, and HFpEF to identify candidate causal variants. The results are summarized in Supplementary Data 12.

#### Proxy and putative functional variants
The prediction scores for non-synonymous substitution of amino acid were summarized as effects on protein (Supplementary Data 13A, B). In addition to the known missense variant (rs2234962) in the BAG3 locus for dilated cardiomyopathy, we identify deleterious or damaging protein-coding variants in genes *SYNPO2L, ERBB2*, and *STARD3* in strong LD with sentinel SNPs (LD $R^2 > 0.8$).

#### Functional annotation of eQTL, pQTL, and enhancers
For unclassified HF, sentinel SNPs rs6795366 and rs34432450 were not found in the database. We used proxy variants passed GWS threshold with strong LD for the search. All sentinel SNPs except rs2634073, rs4977575 and rs79329549 showed evidence of eQTLs in at least one issue type. For HFrEF, all but sentinel SNPs rs2261792, rs56286049, and rs4977575 had significant eQTLs. Identified eQTLs and their tissue types were summarized in Supplementary Data 14. Using the Feland database, we identified 416 pQTLs ($p < 0.0005$) for identified GWAS loci (Supplementary Data 15). We identified 17, 10, and 1 GWS loci overlapping with human enhancers for unclassified HF, HFrEF and HFpEF, respectively (Supplementary Data 16).

#### Genetically predicted gene-expression analysis
Common variants from the different HF subtype GWAS were used to evaluate the association of genetically predicted gene-expression levels with HFrEF and HFpEF across 48 tissues using S-PrediXcan. We identified 49 statistically significant ($P < 5 \times 10^{-7}$) gene-tissue combination pairs genetically predictive of HFrEF risk (Supplementary Data 17), including several gene-expression levels in HFrEF-related tissues such as *CLCNKA* expression in the coronary artery ($5.26 \times 10^{-11}$), *PPP1R1B* ($3.52 \times 10^{-8}$), and *PGAP3* ($1.63 \times 10^{-7}$) expression in left atrial appendage, *PROM1* ($5.57 \times 10^{-8}$), *BPTF* ($9.70 \times 10^{-8}$), and *PGAP3* ($1.44 \times 10^{-7}$) expression in the left ventricle. Hypergeometric enrichment analysis showed that most enriched gene-expression signals (false discovery rate < 0.05) were in three brain tissues, cortex, cervical spinal cord, and substantia nigra. However, we did not identify any genetically predicted gene-expression levels associated with HFpEF.

#### Colocalization analysis
Additionally, we used COLOC to identify the subset of significant genes where there was a high posterior probability that the set of model SNPs in the S-PrediXcan analysis for each gene were both causal for gene expression and HF subtypes. This analysis refined our S-PrediXcan analysis by excluding results that may be the consequence of LD between causal SNPs for gene expression and HF subtypes. All six aforementioned gene-tissue pairs significantly associated with HFrEF has high posterior probability (P4 > 0.9) of colocalization, covering five distinct genes' expression in coronary artery, left atrial appendage and left ventricle.

#### Gene-set and pathway enrichment analysis
To identify pathways and tissues overrepresented in the GWAS of HFrEF and HFpEF, we used the DEPICT gene-set enrichment tool, using all SNPs with *p*-value less than $10^{-4}$ for the respective subtype. We identified four gene sets significantly associated (false discovery rate < 0.05) with HFrEF (Supplementary Data 18) including protein-protein interaction subnetworks. No gene sets were significantly associated with HFpEF using the same approach. We also identified six and six tissue types suggestively associated (false discovery rate < 0.2) with HFrEF and HFpEF, respectively (Supplementary Data 19). The top enriched tissue types including heart and endocrine glands for HFrEF, and blood vessels, epithelial cells, and blood for HFpEF.

### Discussion
In our large-scale genetic association analysis of clinical HF subtypes, we found pronounced differences in the genetic architectures of HFrEF and HFpEF. The very limited genetic discovery in HFpEF in spite of a large cohort size similar to HFrEF, suggests that HFpEF as currently clinically defined is a heterogenous phenotype with varying underlying pathobiology across the phenotype (Fig. 6).

Our genetic analyses of the associations between HF risk factors and HF subtypes, and causal relations of HF risk factors to HFrEF and HFpEF confirmed current epidemiologic data and the validity of our cohorts. For example, we found strong genetic associations of CAD

**Table 3 | Sentinel SNPs significantly associated with HFrEF (19,495 cases) and HFpEF (19,589 cases)**

| rsID | Position | Closest gene | Genomic region | Risk allele/Ref. allele | Risk allele frequency | MVP HFrEF GWAS | | MVP HFpEF GWAS | | HFrEF vs. HFpEF p-value |
|---|---|---|---|---|---|---|---|---|---|---|
| | | | | | | OR (95% CI) | p-value | OR (95% CI) | p-value | |
| **HFrEF** | | | | | | | | | | |
| rs1763610 | 1:16335527 | HSPB7 | Flanking | C/G | 0.64 | 1.11 (1.08, 1.13) | $1.06 \times 10^{-18}$ | 1.00 (0.98, 1.02) | 0.910 | $5.41 \times 10^{-11}$ |
| rs2261792 | 1:61881191 | NFIA | Intron | G/A | 0.36 | 1.06 (1.04, 1.09) | $4.11 \times 10^{-8}$ | 1.00 (0.98, 1.03) | 0.769 | $1.69 \times 10^{-4}$ |
| rs12612948 | 2:11568740 | E2F6 | Flanking | G/A | 0.35 | 1.07 (1.04, 1.09) | $1.27 \times 10^{-8}$ | 1.01 (0.99, 1.03) | 0.407 | $5.99 \times 10^{-4}$ |
| rs56286049 | 3:69824230 | MITF | Intron | C/G | 0.77 | 1.08 (1.05, 1.11) | $7.86 \times 10^{-9}$ | 1.01 (0.98, 1.03) | 0.629 | $6.08 \times 10^{-5}$ |
| rs9349379 | 6:12903957 | PHACTR1 | Intron | G/A | 0.40 | 1.06 (1.04, 1.09) | $5.53 \times 10^{-9}$ | 1.00 (0.98, 1.02) | 0.797 | $5.58 \times 10^{-5}$ |
| rs4151702 | 6:36645988 | CDKN1A | Intron | G/C | 0.79 | 1.15 (1.12, 1.18) | $7.29 \times 10^{-25}$ | 1.01 (0.98, 1.03) | 0.567 | $3.63 \times 10^{-13}$ |
| rs10455872 | 6:161010118 | LPA | Intron | G/A | 0.07 | 1.14 (1.10, 1.19) | $2.17 \times 10^{-11}$ | 1.06 (1.02, 1.11) | $3.87 \times 10^{-3}$ | $9.43 \times 10^{-3}$ |
| rs4977575 | 9:22124744 | CDKN2B-AS | Intergenic | G/C | 0.49 | 1.08 (1.06, 1.11) | $1.80 \times 10^{-13}$ | 1.04 (1.02, 1.06) | $6.62 \times 10^{-4}$ | $3.74 \times 10^{-3}$ |
| rs2234962 | 10:121429633 | BAG3 | Missense | T/C | 0.79 | 1.12 (1.09, 1.15) | $9.02 \times 10^{-18}$ | 0.97 (0.94, 0.99) | $6.42 \times 10^{-3}$ | $1.74 \times 10^{-16}$ |
| rs7306330 | 12:51320290 | METTL7A | Intron | A/T | 0.42 | 1.07 (1.05, 1.09) | $5.58 \times 10^{-10}$ | 1.00 (0.98, 1.02) | 0.996 | $3.98 \times 10^{-6}$ |
| rs7188250 | 16:53834607 | FTO | Intron | C/T | 0.41 | 1.07 (1.04, 1.09) | $2.85 \times 10^{-9}$ | 1.07 (1.05, 1.09) | $9.19 \times 10^{-10}$ | 0.842 |
| rs3764351 | 17:37824339 | PNMT | Intron | G/A | 0.36 | 1.07 (1.05, 1.09) | $4.34 \times 10^{-9}$ | 1.02 (1.00, 1.04) | $6.81 \times 10^{-2}$ | $4.49 \times 10^{-3}$ |
| rs4790908 | 17:65852907 | BPTF | Intron | G/T | 0.20 | 1.08 (1.05, 1.11) | $3.04 \times 10^{-9}$ | 1.04 (1.01, 1.06) | $7.76 \times 10^{-3}$ | 0.017 |
| **HFpEF** | | | | | | | | | | |
| rs11642015 | 16:53802494 | FTO | Intron | T/C | 0.40 | 1.06 (1.04, 1.08) | $7.01 \times 10^{-8}$ | 1.07 (1.05, 1.10) | $6.45 \times 10^{-11}$ | 0.364 |

Chromosomal position is based on GRCh37/hg19 reference. The sentinel SNPs were mapped to the closest refseq genes based on chromosomal base-pair position. All genetic associations were aligned to effects of the risk alleles (i.e., increased risk for HF subtypes).
Ref reference, OR odds ratio, CI confidence interval, GWAS genome-wide association study.

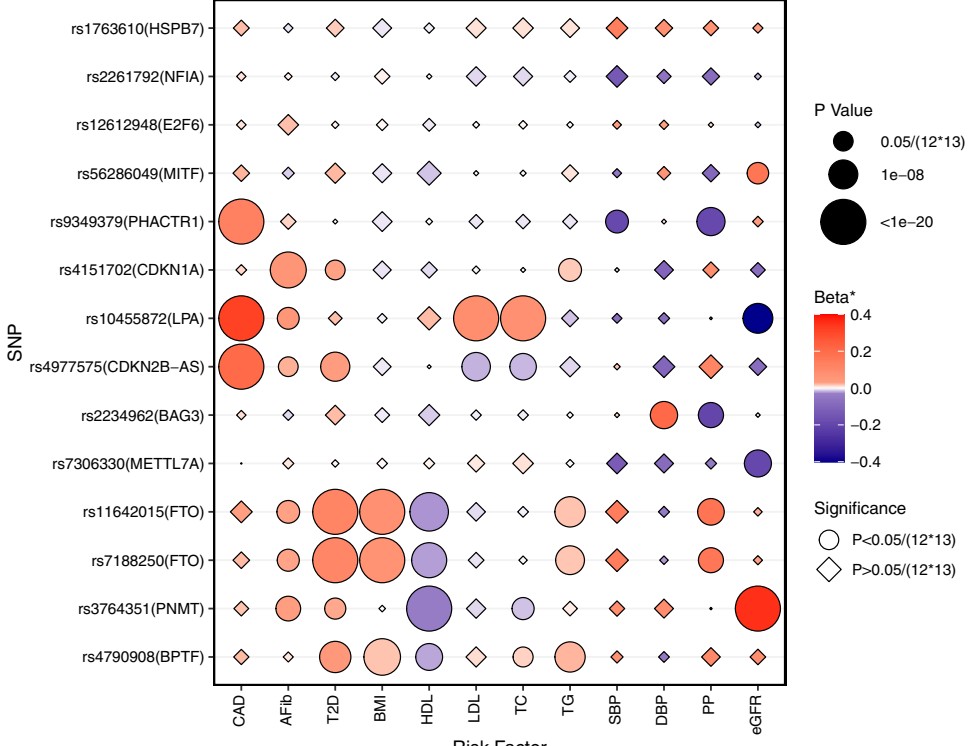

**Fig. 4 | Genetic associations between HFrEF/HFpEF risk variants and HF risk factors.** The genetic associations were identified from published GWAS of HF risk factors. All tests were two-sided without adjustment for multiple comparisons. *Beta: beta coefficients for continuous risk factors, log (odds ratio) for binary risk factors, percent change in eGFR. CAD coronary artery disease, AFib atrial fibrillation, T2D type 2 diabetes, BMI body mass index, HDL high-density lipoprotein cholesterol, LDL low-density lipoprotein cholesterol, TC total cholesterol, TG triglycerides, SBP systolic blood pressure, DBP diastolic blood pressure, PP pulse pressure, eGFR estimated glomerular filtration rate.

and lipid with HFrEF. Conversely, genetically-determined pulse pressure was more associated with HFpEF. Atrial fibrillation and BMI were causally related to both HFrEF and HFpEF. At the level for individual variants, for e.g., in case of the myocardial variant BAG3, different associations were seen with HFpEF and HFrEF. Our finding that the direct genetic correlation between HFrEF and HFpEF was modest (r² ~32%) reinforces our findings at the genomic level that HFrEF and HFpEF have different genetic architecture.

In addition, to ensure that our findings were not due to issues in curating the HFpEF phenotype from the EHR, we used the more restrictive phenotype utilized in our previous epidemiologic studies based on measurement of natriuretic peptides and use of diuretics which had a positive predictive value of 96% on blinded analysis[14] and repeated GWAS in this more restrictively curated sub-group of HFpEF, and found similar genetic associations but less statistical power (due to smaller sample size) comparing to the main HFpEF cohort (Supplementary Fig. 4). Using LDSC and the GWAS summary statistics, we found that the genetic correlation between the two HFpEF definitions was very high (r = 0.981, $p < 2 \times 10^{-16}$). Among top 110 HFpEF-associated common SNPs ($p < 10^{-6}$, MAF > 1%), the genetic effects between the two HFpEF GWAS were highly correlated (r = 0.995, $p < 2 \times 10^{-16}$). Mostly driven by a larger number of HFpEF cases in the original definition (19,598 vs. 12,119), the p-values of 109 out of 110 SNPs were lower in the original HFpEF GWAS conducted in the less restrictive cohort.

The novel genetic associations with HFrEF confirm known pathophysiology and indicate novel biology that merits further investigation. Myocardial remodeling is driven by inappropriate activation of various neurohormonal systems, including the sympathetic nervous system and its effector hormones, the catecholamines epinephrine and norepinephrine[15,16]. Blockade of the adrenergic beta receptor to decrease action of these hormones has substantially

improved survival in HFrEF. The PNMT gene encodes phenylethanolamine N-methyltransferase, which catalyzes the N-methylation of norepinephrine to epinephrine. Sequencing of the PNMT gene has found several SNPs including non-synonymous SNPs in the coding region that affected transcription[17]. Previous studies have associated polymorphisms of the PNMT gene to catecholamine levels and hypertension. Cui and colleagues found that the allelic frequency of an SNP was different between hypertensives and normotensives among African Americans but not among other ethnic groups[18], while Huang et al. found an association of the risk of hypertension with PNMT polymorphisms in Han Chinese population[19]. Polymorphisms of the PNMT gene also influence the levels of post-exercise surge in catecholamine levels[20]. Our data are the first, to our knowledge, that demonstrates an association of PNMT genetic variation with the risk of HFrEF. The gene E2F6 codes for a member of the E2F family of transcription factors that regulate cardiac development, cardiomyocyte growth, and myocardial metabolism[21–23]. Overexpression of E2F6 in the mouse myocardium leads to cardiomyopathy[21], which is associated with decreased glycolytic activity and increased expression of β-hydroxybutyrate dehydrogenase, an enzyme that regulates ketone metabolism[22]. In contrast to the deleterious effects of E2F6 over-expression during cardiac development, in vitro studies have shown that E2F6 may protect against cardiotoxic agents[23]. Preclinical studies have shown that microphthalmia transcription factor (MITF) regulates the hypertrophic response of the myocardium[24], and that the effect of MITF on the myocardial hypertrophic pathway may be mediated epigenetically by the microRNA miR-541[25]. Another potential mechanism of action of MITF on myocardial hypertrophy is via an interaction with four-and-a-half LIM domain protein (FLH2) thereby influencing the expression of ErbB2 interacting protein (Erbin)[26]. While GWAS have shown an association of the PHACTR1 locus with multiple vascular

diseases such as hypertension[27] and coronary calcification[28], down-regulation of PHACTR1 function in vascular cells did not lead to vascular pathology in preclinical studies[29]. The transcription factor NFIA, which has major roles in glial cell development, has been associated with ventricular electrical activity (QRS duration on electrocardiogram) by two population genomic studies[30,31]. In a genetic study of renal and cardiometabolic disease in Zuni Indians, NFIA was associated with diastolic blood pressure[32]. While the function of Methyltransferase Like 7 A (METTL7A) is not well understood, other methyltranferases such as METTL3 and -14 methylate N6-adenosine moieties in RNA and oppose the action of FTO, a N6-adenosine demethylase, which is the only gene that was significantly associated with HF, HFrEF, and HFpEF[33]; myocardial changes in N6-adenosine methylation of mRNA is associated with progression to HF[34].

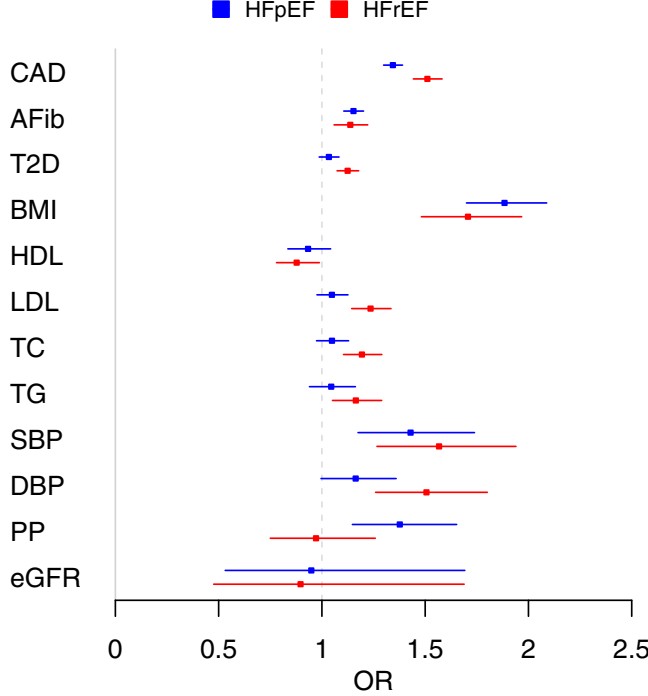

**Fig. 5 | Mendelian randomization analysis of HF risk factors in relation to HFpEF and HFrEF.** X-axis shows odds ratios (ORs), with error bars showing 95% confidence intervals. CAD coronary artery disease, AFib atrial fibrillation, T2D type 2 diabetes, BMI body mass index, HDL high-density lipoprotein cholesterol, LDL low-density lipoprotein cholesterol, TC total cholesterol, TG triglycerides, SBP systolic blood pressure, DBP diastolic blood pressure, PP pulse pressure, eGFR estimated glomerular filtration rate.

There is a developing consensus that HFpEF as currently defined may not represent a cohesive pathophysiology, rather, that HFpEF represents a heterogenous entity comprised of multiple phenotypes. Multiple large randomized clinical trials utilizing medications that were found to be effective in preclinical models of HFpEF did not demonstrate beneficial effects[4]. The contrast with HFrEF is a major reason to conclude that HFpEF may be a heterogenous disease. Although both HFpEF and HFrEF are associated with risk factors and comorbidities, animal models of HFrEF have identified drug targets which have been conclusively proven to reduce morbidity and mortality by large clinical trials[15]. This is in stark contrast to HFpEF, in which the animal models while recapitulating the cardiac pathophysiology, have failed to identify drug targets that benefit HFpEF, suggesting that the pathophysiology of HFpEF may not be as uniform as seen in HFrEF. Our study also showed that despite increased phenotypic refinement of unclassified HF into HFpEF and HFrEF cohorts of similar size, the yield of GWS loci in HFpEF was even lower than in unclassified HF and in contrast to the increased genetic discovery in the HFrEF cohort. We recognize that this does not directly translate into a conclusion of pathophysiologic heterogeneity since many factors influence the pathway from genotype to phenotype, and it is also possible that appropriate drug targets have yet to be identified for HFpEF; however, these findings do suggest that pathophysiologic heterogeneity may have played a significant role in our findings. Our findings suggest an urgent need to develop consensus subphenotyping strategies to resolve the heterogeneity of HFpEF as currently defined, as will be the focus of the recently initiated National Institutes of Health HeartShare Program (https://grants.nih.gov/grants/guide/rfa-files/RFA-HL-21-015.html).

Initial studies that applied unsupervised clustering approaches to clinical and biomarker data mainly derived from HFpEF clinical trials suggest that different subphenotypes may underlie HFpEF[35–38]. For example, Cohen and colleagues used latent class analysis on data from the TOPCAT Trial (Treatment of Preserved Cardiac Function Heart Failure with an Aldosterone Antagonist Trial) and identified three subphenotypes of HFpEF, with one of the subphenotypes associated with better response to spironolactone[36]. Based on these initial results, it is possible that artificial intelligence/machine learning approaches applied to clinical, imaging, biomarker, and -omics data may identify specific subphenotypes of HFpEF that may be benefited by specific drug therapy. While artificial intelligence/machine learning approaches applied to clinical and biomarker data may resolve some of the heterogeneity of HFpEF, biologically based approaches to address the potential for rare genetic variants to influence disease pathogenesis and the complexity of the path from genotype to phenotype using multi-omics, epigenomics and chromatin dynamics, and single cell approaches, may be needed to truly uncover the pathobiology of HFpEF.

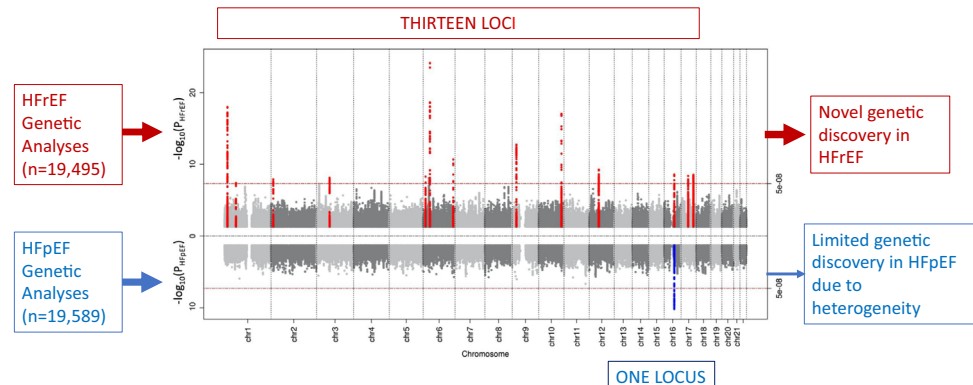

**Fig. 6 | Limited genetic discovery in HFpEF due to pathophysiological heterogeneity.** All tests were two-sided without adjustment for multiple comparisons.

## Study limitations

Our findings should be interpreted in the context of the strengths and limitations of the study. Our HFpEF cohort had less women compared to epidemiologic studies and recent clinical trials; however, the genetic and causal associations of risk factors with HFpEF as compared to HFrEF mirrored associations seen in epidemiologic studies. Since we utilized natural language processing to capture all recorded LVEFs including measurements performed outside the VA, our cohort of HFpEF excluded any participants with previously reduced and currently normal LVEF. In addition, we compared GWAS findings between a more restrictive HFpEF phenotype and the less restrictive phenotype used in the main analysis and found very high correlation confirming the validity of the HFpEF phenotype used for the main GWAS. Hence our findings indicate that the issue with reduced genetic discovery in our cohort was not secondary to impurity of the phenotype due to EHR-based curation, but that HFpEF as currently defined may be a collection of subphenotypes with multiple independent disease mechanisms. Our case and control cohorts, since they were recruited from a hospital setting, had a higher prevalence of comorbidities compared to a population-based cohort. We could not externally replicate our findings since currently there are no other large phenotyped cohorts of HFpEF and HFrEF.

In conclusion, the genetic architectures of HFpEF and HFrEF differ significantly. HFpEF as currently clinically defined is a pathophysiologically heterogenous disease that requires further characterization into consensus subphenotypes to enhance genetic discovery. Better genetic understanding of HF subtypes will lead to precise diagnosis, accurate risk assessment, and effective treatment and management of the global pandemic of heart failure.

## Methods

All research procedures complies with all relevant ethical regulations and were approved by the Institutional Review Boards of Atlanta VA Medical Center and VA Boston Healthcare System.

### Datasets

**Million Veteran Program.** The design of MVP has been previously described[39]. Veterans were recruited from over 60 Veterans Health Administration medical centers nationwide since 2011. A unique feature of MVP is the linkage of a large biobank to an extensive, national, database from 2003 onward that integrates multiple elements such as diagnosis codes, procedure codes, laboratory values, and imaging reports, which permits detailed phenotyping of this large cohort. MVP has received ethical and study protocol approval by the Veterans Affairs Central Institutional Review Board in accordance with the principles outlined in the Declaration of Helsinki.

**UK Biobank.** UK Biobank is a prospective study with over 500,000 participants aged 40–69 years recruited in 2006–2010 with extensive phenotypic and genotypic data[40].

### Phenotyping of heart failure, HFrEF, and HFpEF

HF patients were identified as those with an International Classification of Diseases (ICD)-9 code of 428.x or ICD-10 code of I50.x and an echocardiogram performed within 6 months of diagnosis (median time period from diagnosis to echocardiography was 3 days, interquartile range 0–32 days). Since the accurate classification of HF into HFrEF and HFpEF is dependent on capture of LVEF values, we used a comprehensive approach based on natural language processing (NLP). As previously described, an NLP tool was developed and validated in the national VA database to extract LVEF values from echocardiogram reports[41]. We utilized NLP to capture LVEF values from nuclear medicine reports, cardiac catheterization reports, history and physical examination notes, progress notes,

discharge summary notes, and other cardiology notes, to ensure that we captured LVEF values measured outside the VA[42]. Using analysis of patient records by blinded physician reviewers, we validated the accuracy of the NLP algorithms to capture LVEF and correctly classify HFpEF[42]. Compared to our previous studies, we utilized a wider time frame between HF diagnosis and first recorded LVEF for this study to ensure that we captured LVEFs recorded outside the VA soon after HF diagnosis but entered into the VA medical records later. We classified HFpEF as presence of HF diagnostic code and first recorded EF of ≥50% and HFrEF as HF diagnostic code with first recorded LVEF of ≤40%.

Our HF phenotyping algorithms utilize both structured and unstructured data to ensure accuracy of the HF diagnosis, and natural language processing to ascertain all measurements of left ventricular function from imaging studies (i.e., echocardiograms) and from clinical notes, with the latter permitting capture of left ventricular ejection fractions (LVEF) measured outside the VA system[14,42,43]. Capture of all LVEFs ensured that we truly obtained the LVEF measured at the time of diagnosis of HF to allow proper identification of HFpEF and exclude any veteran with recovered LVEF from the HFpEF cohort. In the algorithm for identification of HF patients, we used documentation in EHR of the ordering of B-type natriuretic peptide as one of the criteria, since evaluation of practice patterns indicated that ordering of B-type natriuretic peptide increased the likelihood of the patient having clinical HF, as validated by blinded review[42]. For this study, to increase the number of HFpEF patients included in the study, we utilized a less restrictive definition recently utilized in a study[44] that did not require that all LVEFs recorded after the baseline measurement also be ≥50%, or the use of diuretics and/or measurement of B-type natriuretic peptide at baseline (Fig. 2). To ensure adequacy of this definition, we compared the genetic associations obtained in the cohort to genetic associations obtained in a cohort curated with the more restrictive definition used for our previous epidemiological studies[14,41,42,45]. Comorbid conditions were curated using International Classification of Diseases (ICD)-10 or ICD-9 codes as in our previous studies and described in the Supplementary Materials[42].

In the UK Biobank, we defined HF as the presence of self-reported HF/pulmonary edema or cardiomyopathy at any visit; or an ICD-10 or ICD-9 billing code indicative of heart/ventricular failure or a cardiomyopathy of any cause, as described and validated previously, and consistent with that used in a recent, international collaborative effort[8,46] Assessments of LVEF were not available in the majority of UK Biobank participants to permit classification into HFpEF and HFrEF.

### Genetic data production, quality control, and imputation

DNA extracted from participants' blood was genotyped using a customized Affymetrix Axiom® biobank array, the MVP 1.0 Genotyping Array. The array was enriched for both common and rare genetic variants of clinical significance in different ethnic backgrounds. Quality-control procedures used to assign ancestry, remove low-quality samples and variants, and perform genotype imputation were previously described[47]. We excluded: duplicate samples, samples with more heterozygosity than expected, an excess (>2.5%) of missing genotype calls, or discordance between genetically inferred sex and phenotypic gender[47]. In addition, one individual from each pair of related individuals (more than second degree relatedness as measured by the KING software)[48] were removed. Prior to imputation, variants that were poorly called (genotype missingness > 5%) or that deviated from their expected allele frequency observed in the 1000 Genomes reference data were excluded. After pre-phasing using EAGLE v2.4[49], we then imputed to the 1000 Genomes phase 3 version 5 reference panel (1000 G) using Minimac4[50]. Genotyped SNPs after quality control were interpolated into the imputation file. Imputed variants with poor imputation quality ($r^2 < 0.3$) were excluded from further analyses.

## Assignment of racial/ethnic groups in the MVP

The MVP participants were assigned to mutually exclusive racial/ethnic groups using HARE (Harmonized Ancestry and Race/Ethnicity), a machine learning algorithm that integrates genetically inferred ancestry (GIA) with self-identified race/ethnicity (SIRE)[51]. HARE defines ethnicity-specific strata by a two-step process: an initial training step in which a support vector machine model was built and made to learn the correspondence between genetically inferred ancestry (GIA) and SIRE; and a second assignment step in which HARE was derived from SIRE, GIA, and the output from the support vector machine.

## Genome-wide association analysis

Figure 1 demonstrates our study schema. Imputed and directly measured single nucleotide polymorphisms (SNPs) with minor allele frequency >1% were tested for association with HF, HFrEF, and HFpEF assuming an additive genetic model using PLINK2[52] and adjusting for age, sex, and the top ten genotype-derived principal components. In UK Biobank analyses, genotyping array was included as an additional covariate. We meta-analyzed GWAS results of HF from MVP and UK Biobank using inverse-variance weighted fixed-effects model implemented in METAL[53]. Joint meta-analysis results were reported for unclassified HF to improve the power for GWAS discovery[54]. GWAS results were summarized using FUMA, a platform that annotates, prioritizes, visualizes and interprets GWAS results[55]. Genome-wide significant SNPs ($P < 5 \times 10^{-8}$) were grouped into a genomic locus based on either $r^2 > 0.1$ or distance between loci of <500 kb using the 1000 Genomes European reference panel. Lead SNPs were defined within each locus if they were independent ($r^2 < 0.1$). We considered loci as novel if the sentinel SNP was of genome-wide significance ($P < 5 \times 10^{-8}$) and located >1 Mb from previously reported GWS SNPs associated with HF[8,46]. For novel loci, we used the genomic base-pair position of each sentinel SNP to map to the closest gene within a 500 kb region as the candidate gene. The physical base-pair location (GRCh37/hg19) and alleles were used to uniquely identify a genetic variant to replicate previous reported genetic associations with HF, and with HF risk factors.

For replication of unclassified HF, we conducted genome-wide association testing among UK Biobank participants passing sample quality control, comparing unclassified HF cases with non-HF controls. Procedures for genotyping and genotype imputation in the UK Biobank have been described previously[40]. For genetic association testing, we included SNPs with minor allele frequency (MAF) > 1% available in the Haplotype Reference Consortium (HRC), and imputation quality (INFO) > 0.3. We restricted analyses to samples of European genetic ancestry, defined by a combination of self-reported race and genetic principal components of ancestry. Specifically, we selected samples with genetic data who self-reported as white (British, Irish, or Other) and applied an outlier detection protocol (R package aberrant) to three pairs of principal components (PC1/PC2, PC3/PC4, and PC5/PC6), as generated centrally by the UK Biobank. Outliers in any of the three pairs of PCs were excluded from analysis to ensure that the study population was relatively homogenous in terms of genetic ancestry. Additional sample exclusions were implemented for 2nd-degree or closer relatedness (Kinship coefficient > 0.0884), sex chromosome aneuploidy, and excess missingness or heterozygosity, as defined by the UK Biobank. Association analyses were performed using PLINK2 (https://www.cog-genomics.org/plink/2.0/) 25 on imputed genotype dosages, and a logistic regression model was used adjusting for age at enrollment, sex, genotyping array, and the first 10 principal components of ancestry. After merging with the phenotypic data, a total of 8227 unclassified HF cases were compared to 379,788 non-HF controls. Test statistic inflation was investigated by genomic control and inspection of quantile-quantile plots.

## Genetic correlation and heritability

We estimated genetic correlations between these complex traits using cross-trait LD Score Regression and European ancestry-based GWAS

results of HFpEF and HFrEF[56,57]. A reference panel consisting of 1.2 million HapMap3 variants was used to merge with GWAS summary statistics filtered to variants with MAF > 0.01, Hardy-Weinberg equilibrium $P > 10^{-20}$ and imputation $R^2 > 0.5$. Using LD Score Regression and GWAS summary statistics, we also estimated the inflation factor of unclassified HF, HFpEF and HFrEF.

We used GREML-LDMS-I as implemented in Genome-wide Complex Trait Analysis (GCTA) 1.93.0beta to estimate the multicomponent heritability of unclassified HF, HFrEF, and HFpEF in our MVP participants of European ancestry. GREML-LDMS-I was shown to be the least biased and one of the most accurate heritability estimation methods[58]. Restricted by computing memory requirements, we randomly selected 50,000 unrelated MVP non-Hispanic Whites to perform GREML-LDMS-I analysis[59,60]. We then estimated heritability within each group after applying identical quality-control procedures. SNPs that were multi-allelic, had MAC < 6, or call-rate <95% were removed. LD scores were computed on each autosome using an $r^2$ cutoff of 0.01, and the genome-wide LD score distribution was used to assign SNPs to 1 of 4 LD quartile groups, where groups 1–4 represented SNPs with higher LD scores. Within each LD group, SNPs were further stratified into 6 MAF bins ([0.001, 0.01], [0.01, 0.1], [0.1, 0.2], [0.2, 0.3], [0.3, 0.4], [0.4, 0.5]) and a genetic relatedness matrix (GRM) was constructed from each bin, creating 24 GRMs. Finally, GCTA -reml was used to fit a model of HF case status based on the 24 GRMs, with age and sex as covariates. Total observed heritability estimates were transformed to estimate disease liability scale across a range of presumed HF subtypes prevalence estimates (2.5% to 7% for each HF subtype).

## Mendelian randomization analysis of HF risk factors

To assess differential causal associations of risk factors with HFrEF and HFpEF, we conducted two-sample Mendelian Randomization (MR). For MR, we utilized genetic instrumental variables reported in previous GWAS of the following traditional HF risk factors: coronary artery disease (CAD)[61], atrial fibrillation (AF)[62], type 2 diabetes (T2D)[63], body mass index (BMI)[64], lipids[65], blood pressure[66], and estimated glomerular filtration rate (eGFR)[67]. The GWS sentinel SNPs from each GWAS were selected as the genetic instrumental variables (GIVs) for each HF risk factor. We estimated the MR association of each risk factor using three complementary methods: inverse-variance-weighted, median weighted, and MR-Egger regression, as implemented in the R package TwoSampleMR[68]. MR-Egger regression was used to identify the horizontal pleiotropy measured by the intercept of the regression. Random-effects model was used to estimate the MR association between HF risk factors and HF outcomes for IVW and MR-Egger regression. To avoid sample overlap in the two-sample MR design, we used summary statistics of unclassified HF, HFrEF, and HFpEF from the MVP study, and summary statistics of risk factors in previous GWAS without the MVP, all from studies of European ancestry. We considered nominal p-value of 0.05 as suggestive evidence for MR association for each HF risk factor. We applied a stringent Bonferroni correction for 12 tested factors (p-value < 0.05/12 = 0.0042) acknowledging that some factors are not independent.

## Conditional analysis and credible set analysis

To determine the presence of independent secondary signals within the GWS loci of HF and subtypes, we conducted a conditional analysis using -cojo-cond command implemented in the genome-wide complex trait analysis (GCTA) tool. A secondary independent signal is defined as a SNP with the conditional p-value less than $5 \times 10^{-8}$ within a ± 500 kb flanking region of the sentinel SNP of each identified locus.

We generated a list of credible sets of SNPs at all GWS loci of unclassified HF, HFrEF, and HFpEF in European ancestry using a Bayesian approach for credible set analysis[69]. We first calculated approximate Bayes factors for each variant within a 500 kb region

centered on the sentinel SNP using the beta, standard error, and sample size from the METAL meta-analysis of unclassified HF and the MVP GWAS of HFrEF and HFpEF. We then estimated the posterior probability of each variant being causal using the Bayesian factor. Lastly, a credible set was defined as the smallest set of SNPs for which the sum of posterior probability reached 95%.

### Proxy and putative functional variants

For each region, we explored the effect of non-synonymous coding SNPs on protein function using the variant annotation tool SNPnexus (https://www.snp-nexus.org/v4/), including molecular function and polymorphism phenotyping predictions from SIFT[70] and PolyPhen[71], within a 500 kb region centered around the sentinel SNPs[72].

### Functional annotation of eQTL, pQTL, and enhancers

Using GTEx database including a set of 49 tissues, we searched for the eQTLs for the genetic variants associated with unclassified HF and its two subtypes at $p < 0.0005$. We obtained protein-quantitative trait loci (pQTLs) from the Fenland study, a genome-proteome-wide association study in 10,708 European-descent individuals. The genome-proteome-wide association study was performed using 10.2 million genetic variants including plasma abundances of 4775 distinct protein targets measured using the SOMAscan V4 assay in plasma[73]. The SOMAscan assay employs single-stranded oligonucleotides (aptamers) with specific binding affinity to a single protein. We retrieved functional annotations from the Fenland proteo-genomic study for each SNP we identified for unclassified HF, HFpEF, and HFrEF, matched by chromosomal position and reference allele ($p < 0.0005$). We also searched 193,218 enhancers regions from 295 cell/tissue types from EnhancerAtlas[74] for all identified sentinel SNPs.

### Genetically predicted gene-expression analysis

Genetically predicted gene expression was estimated using S-PrediXcan, an approach that imputes genetically predicted gene expression (GPGE) in a given tissue and tests predicted expression for association with a trait using GWAS summary statistics. For this analysis, input included results for common variants in our heart failure GWAS and gene-expression references for 48 tissues from GTEx[75]. Our analyses incorporated covariance matrices based on the 1000 Genomes Project European populations to account for LD structure[76]. Bonferroni-corrected significance threshold was $1.93 \times 10^{-7}$ for these analyses.

### Colocalization analysis

The hypothesis that a single variant underlies GWAS and expression quantitative trait loci (eQTL) associations at a given locus (i.e., colocalization) was tested using coloc[77], a gene-level Bayesian test that evaluates GWAS and eQTL association summary statistics at each SNP at the locus and provides gene- and SNP-level posterior probabilities for colocalization. For this analysis, input included results for common variants in our GWAS and eQTL summary statistics corresponding to the gene-expression references used in S-PrediXcan analysis.

### Gene-set and pathway enrichment analysis

Gene-set and pathway enrichment analysis was performed using DEPICT for HFrEF and HFpEF with both genome-wide significant SNPs ($p < 5 \times 10^{-8}$) and suggestive signals using a less stringent threshold ($p < 10^{-4}$)[78]. Common SNPs with MAF > 0.01, HWE $p > 10^{-20}$ and imputation $R^2 > 0.5$ were included in the analysis.

### Reporting summary

Further information on research design is available in the Nature Portfolio Reporting Summary linked to this article.

## Data availability

Due to US Department of Veterans Affairs (VA) regulations and our ethics agreements, the analytic datasets used for this study are not permitted to leave the Million Veteran Program (MVP) research environment and VA firewall. This limitation is consistent with other MVP studies based on VA data. However, the MVP data are made available to researchers with an approved VA and MVP study protocol. The full summary level association data genome-wide association analyses in the MVP and the meta-analysis from this report will be available through dbGaP (accession number phs001672). The only restriction is that use of the data is limited to health/medical/biomedical purposes, and does not include the study of population origins or ancestry. Use of the data does include methods development research (e.g., development and testing of software or algorithms) and requestors agree to make the results of studies using the data available to the larger scientific community. We used publicly available data from GTEx (https://gtexportal.org/home/).

## Code availability

We utilized publicly available software for all analyses, and software used in this study is described in the Methods section.

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

## Acknowledgements

We are grateful to all the MVP investigators; a list of MVP investigators can be found in Supplementary Materials. This research is supported by funding from the Department of Veterans Affairs Office of Research and Development, Million Veteran Program Grant I01-CX001737 (PI: Phillips) and I01-BX004821 (PI: Wilson). This publication does not represent the views of the Department of Veterans Affairs or the United States Government.

## Author contributions

J.J. and Y.V.S. conceived of the project, oversaw the analyses and interpretation, and collaborated on writing and finalizing the manuscript. Q.H., C.L., K.A., Z.W., J.K., T.E., and B.C. performed the analyses and participated in the writing of the manuscript. K.A., S.D., L.D., J.H., J.P.C., J.M.G., K.C., P.W.F.W., L.S.P., and C.O.D. participated in the conception of study and analyses of data, and in the writing of the manuscript.

## Competing interests

The authors declare no competing interests.

## Additional information

# VA Million Veteran Program

Jacob Joseph [1,2,3,13] ✉, Jennifer E. Huffman[1,2], Luc Djousse[1,2], Juan P. Casas[1,2], J. Michael Gaziano[1,2], Kelly Cho[1,2], Peter W. F. Wilson[5,12], Lawrence S. Phillips[5,12] & Yan V. Sun [4,5,13] ✉

A full list of members and their affiliations appears in the Supplementary Information.

