## [Peer Review File · Nature Communications]

Genetic Architecture of Heart Failure with Preserved versus Reduced Ejection FractionREVIEWER COMMENTS

Reviewer #1 (Remarks to the Author):

GWAS for heart failure (HF) is recognized as challenging because of the small number of genome-wide significant loci identified for a large same sample size relative to other common cardiovascular diseases such as coronary artery disease and atrial fibrillation. While GWASs for cardiac function measures such as echocardiography and cardiac MRI have led to different approaches to heart failure in recent years, this is the first study to attempt to determine the genetic background of HFrEF (HF reduced ejection fraction) and HFpEF (HF preserved ejection fraction). The authors performed GWAS for all-cause HF, HFrEF, and HFpEF, followed by revealing their commonalities and heterogeneities. They have concluded that there should be subcategories within HFpEF and that the development of the classification approach is urgently needed, since the number of significant loci for HFrEF and HFpEF differed significantly despite the similar sample sizes.

The heterogeneity between HFrEF and HFpEF is of great clinical interest, and this study, which tackled that topic with a large sample, is of great value. On the other hand, I have several concerns.

Major comments

Overall, I feel that the interpretation of genome-wide significant loci identified was discussed only in terms of nearby genes and known clinical risk factors, and the molecular biology and comprehensive approaches were somewhat lacking.

1. I think it is better to perform conditional analysis on identified loci to determine how many independent signals were present.
2. Also, fine-mapping should be performed to point out causal variants in a locus. Because this study is mainly based on GWAS for Europeans, fine-mapping using LD information should be easy.
3. Are there any protein-coding variants that exist in the same LD of sentinel SNPs? That could suggest functionally associated genes.
4. Regarding putative causal variants which could be identified by fine-mapping, can you have any interpretations related to biological mechanisms such as eQTLs and pQTLs? and what about epigenetic states (e.g., enhancer regions)? Can you suggest putative causal genes by TWAS, etc.?
5. Disease-prevalent tissues/cells should be detected by using S-LDSC or DEPICT.

6. Line 111-112.

The authors mentioned the difference in the effect direction of sentinel SNPs between HFpEF and HFrEF, What statistical model was used to test this? To show the difference, the Heterogeneity P-value obtained from a fixed-effect meta-analysis could be useful. I also propose that a figure could be intuitive where the value of beta in HFpEF and HFrEF are plotted on the X and Y axes, respectively, and where the dissociation from $Y = X$ is shown.

7. Line 113-115.

BAG3 is a famous pathogenic gene for dilated cardiomyopathy. So it was surprising that BAG3 missense variant (rs2234962) was associated with lower risk for HFpEF. Could you describe a possible reason for this?

8. Line 135. "15 autosomal genes related to cardiomyopathy."

Reference 14 is somewhat outdated. The latest one is available (<https://doi.org/10.1038/s41436-018-0039-z>). According to it, there may be more than 15 genes.

9. Line 144-151. Associations of HFrEF- and HFpEF Loci with Cardiovascular Risk Factors

I wonder if the summary statistics of UKBB published by Ben Neal's lab would be better to compare without bias caused by study differences. Also, is it possible to look at associations comprehensively without limiting them to cardiovascular risk factors?

Besides, it seems that one variant was determined to see the association. I think colocalization

analysis should be performed to determine the association.

10. Lines 159-168. Mendelian Randomization Association Analysis of HF Risk Factors

Like 8, I wonder if it would be better to select instrumental variables from summary statistics of the UKBB Ben neal lab provided, instead of taking it from a previously published GWAS study with a different background, so that the background of the study does not become a bias and we can explore more traits. I also suppose it would be interesting to consider both cases "outcome is HF" and "Exposure is HF".

11. Lines 173-175, 183-185.

The authors speculated that the reason why there were fewer significant loci in HFpEF compared to HFrfEF was due to the Heterogeneity of HFpEF. And, as an example, they cited a stroke GWAS. However, I have not been convinced by this example. Please explain in more detail the rationale for assuming that heterogeneity within the disease, like stroke, is the cause of the fewer loci in HFpEF. It may simply be that the heritability is small due to large environmental factors.

12. The authors utilized natural language processing to exclude HF with improved EF. I think the validation of natural language processing is required (eg. compare results obtained in and outside VA).

Minor comments

1. The number of cases differed in figure 2.

Line 81. HFpEF case = 19,589.

In figure 1, HFpEF case = 19,589.

In figure 2, HFpEF case = 19,598.

2. Line 97. "established definitions of HFrfEF and HFpEF"

The definition of HFpEF and HFrfEF using the EF threshold is often used and prevalent, but I do not think "established".

3. Line 114, 132.

Is this just because HFpEF and HFrfEF are competing risks to each other?

4. Line 120-121.

It would be easier to understand if there is also a "ratio" value obtained from LDSC to show the inflation-polygenicity ratio.

5. Line.153-157. Heritability & Genetic Correlation

Calculation with LDSC is based on summary statistics. Since raw data is available, would there be any difference if it is calculated with GREML?

6. Lines 210-213. "Hence our findings indicate that the issue with reduced genetic discovery in our cohort was not secondary to the impurity of the phenotype due to EMR-based curation, but rather that HFpEF as currently defined may be a collection of sub-phenotypes with multiple independent disease mechanisms."

It would be appreciated if there are proposals on how to do this as well as the need for subclassification for HFpEF. Any such hints from clinical studies? What about from the genome side?

7. Line 212.

EMR (electronic medical records?) should be spelled out because it was not defined anywhere in this manuscript.

8. Line 247. "measurement of B-type natriuretic peptide at baseline"

It is not essential to measure BNP to diagnose HF. A high BNP value should be a clue. So, please

explain why just measuring BNP can be a criterion to diagnose HF.

9. Line. 264-266. " We excluded: duplicate samples, samples with more heterozygosity than expected, an excess (>2.5%) of missing genotype calls, or discordance between genetically inferred sex and phenotypic gender."

Since the association tests were performed based on a logistic regression model implemented in PLINK2, related individuals in addition to duplicated ones should be excluded?

10. Whilst the main theme of this study is the genetic difference between HFpEF and HFrEF, I felt that the consideration and comparison with African Americans were insufficient.

11. Figure 1 and Figure 2, and Tables S4 and S5 are in the opposite order compared to the main manuscript.

Reviewer #2 (Remarks to the Author):

The authors describe a GWAS study of heart failure (HF) comparing reduced (HFrEF) versus preserved (HFpEF) ejection fraction. This is an interesting study. A genetic comparison of these two clinical subtypes of heart failure is an important and not insignificant undertaking.

It is strongly argued the pathological heterogeneity of HFpEF explains the largely failed outcomes of clinical trials. The authors believe clinically defined HFpEF represents an amalgamation of several different pathobiological entities and highlight the need to sub-phenotype it. Even with machine learning, it is not clear how this might be easily accomplished, and there could be multiple reasons for failed clinical trials, including lack of suitable targets!

This is an impressive GWAS, but have quantitative GWAS approaches been considered? The samples provided by the Million Veteran Program appear to have been well phenotyped, but the description of phenotyping is vague. Notwithstanding this, the authors replicated all 12 SNPs associated with HF, previously published in Nat Commun (Shah et al., 2020). The associations with unclassified HF identified in patients of European ancestry were then replicated in African Americans. The authors follow up with a Mendelian randomisation study linking well-known risk factors with HFrEF and HFpEF.

The authors casually mention using the less restrictive definitions for HFpEF because it is better powered, but this needs a stronger justification. The authors could also have discussed the heritability (h^2) scores.

Overall, this is a good piece of work, but the potential biological relevance of the loci identified in relation to this and related studies is hardly mentioned. Apart from highlighting the sample size, the authors don't do themselves any justice, by failing to highlight the novelty of this work, and its implications. Perhaps, pathway analysis might have been informative.

Clearly, like all other complex diseases, HF is clinically and genetically heterogeneous. However, the focus for genetic studies of HF (and other complex diseases) is shifting, with increasing attention on rare variant analysis, chromatin dynamics, and single cell approaches, which are more likely to deliver new insights than bigger GWASs.

REVIEWER COMMENTS

We are grateful to the reviewers for their comments and have done additional work and revised the manuscript as suggested. Point-by-point response is detailed below. All changes in the revised manuscript are highlighted.

Reviewer #1 (Remarks to the Author):

GWAS for heart failure (HF) is recognized as challenging because of the small number of genome-wide significant loci identified for a large same sample size relative to other common cardiovascular diseases such as coronary artery disease and atrial fibrillation. While GWASs for cardiac function measures such as echocardiography and cardiac MRI have led to different approaches to heart failure in recent years, this is the first study to attempt to determine the genetic background of HFrEF (HF reduced ejection fraction) and HFpEF (HF preserved ejection fraction). The authors performed GWAS for all-cause HF, HFrEF, and HFpEF, followed by revealing their commonalities and heterogeneities. They have concluded that there should be subcategories within HFpEF and that the development of the classification approach is urgently needed, since the number of significant loci for HFrEF and HFpEF differed significantly despite the similar sample sizes.

The heterogeneity between HFrEF and HFpEF is of great clinical interest, and this study, which tackled that topic with a large sample, is of great value. On the other hand, I have several concerns.

We appreciate reviewer's recognition of the major strengths of our study. We provide point-by-point response to address all raised concerns in the following section.

Major comments

Overall, I feel that the interpretation of genome-wide significant loci identified was discussed only in terms of nearby genes and known clinical risk factors, and the molecular biology and comprehensive approaches were somewhat lacking.

1. I think it is better to perform conditional analysis on identified loci to determine how many independent signals were present.

We performed the conditional analysis using the COJO function implemented in the GCTA software. We identified two additional GWS SNPs conditional on the sentinel SNPs of unclassified HF and updated the results including Supplementary Tables 11.

Methods

“Conditional analysis

To determine the presence of independent secondary signals within the GWS loci of HF and subtypes, we conducted a conditional analysis using --cojo-cond command implemented in the genome-wide complex trait analysis (GCTA) tool. A secondary independent signal is defined as a SNP with the conditional p-value less than 5×10^{-8} within a ± 500 kb flanking region of the sentinel SNP of each identified locus.”

Results

“We identified a secondary SNP in two loci on chromosome 4 and 6 after conditional analysis on the sentinel SNP from unclassified HF GWAS (Supplementary Table 11). However, there was no evidence of secondary independent variants at any GWS loci of HF subtypes in conditional analyses.”

2. Also, fine-mapping should be performed to point out causal variants in a locus. Because this study is mainly based on GWAS for Europeans, fine-mapping using LD information should be easy.

To identify potential causal variants of identified loci of unclassified HF, HFrEF and HFpEF, we performed credible set and colocalization analyses considering the local LD structure and functional annotations. We included more details for the credible set analysis (see below) and colocalization (see major comment #4) in the revised manuscript.

Methods

“Credible set analysis

We generated a list of credible sets of SNPs at all GWS loci of unclassified HF, HFrEF and HFpEF in European ancestry using a Bayesian approach for credible set analysis (PMID: 32282791). We first calculated approximate Bayes factors for each variant within a 500 kb region centered on the sentinel SNP using the beta, standard error, and sample size from the METAL meta-analysis of unclassified HF and the MVP GWAS of HFrEF and HFpEF. We then estimated the posterior probability of each variant being causal using the Bayesian factor. Lastly, a credible set was defined as the smallest set of SNPs for which the sum of posterior probability reached 95%.”

Results

“We performed a credible set analysis of all GWS loci for unclassified HF, HFrEF and HFpEF to identify candidate causal variants. The results are summarized in Supplementary Table 12.”

3. Are there any protein-coding variants that exist in the same LD of sentinel SNPs? That could suggest functionally associated genes.

We identified proxy and putative functional variants within a 500 kb region centered around the sentinel SNPs associated with unclassified HF and the two clinical subtypes. In addition to the known missense variant in the *BAG3* locus for dilated cardiomyopathy, we identify deleterious or damaging protein-coding variants in genes *SYNPO2L*, *ERBB2* and *STARD3* in strong LD with sentinel SNPs ($LD R^2 > 0.8$). We revised the Methods and Results sections as the following.

Methods

“For each region, we explored the effect of non-synonymous coding SNPs on protein function using the variant annotation tool SNPnexus (<https://www.snp-nexus.org/v4/>), including molecular function and polymorphism phenotyping predictions from SIFT (PMID: 19561590) and PolyPhen (PMID: 23315928), within a 500 kb region centered around the sentinel SNPs (PMID: 32180801).”

Results

“The prediction scores for non-synonymous substitution of amino acid were summarized as effects on protein (Supplementary Tables 13A and 13B). In addition to the known missense variant (rs2234962) in the *BAG3* locus for dilated cardiomyopathy, we identify deleterious or damaging protein-coding variants in genes *SYNPO2L*, *ERBB2* and *STARD3* in strong LD with sentinel SNPs ($LD R^2 > 0.8$).”

4. Regarding putative causal variants which could be identified by fine-mapping, can you have any interpretations related to biological mechanisms such as eQTLs and pQTLs? and what about epigenetic states (e.g., enhancer regions)? Can you suggest putative causal genes by TWAS, etc.?

We appreciate the reviewer’s suggestions of linking identified loci to molecular mechanism using gene expression, proteomics and epigenetics/enhancer data. We included additional functional annotation by searching for available databases of eQTL, pQTL and enhancers to provide evidence of putative molecular functions of GWS loci. The complete search results were included in additional supplementary tables. We also included TWAS/colocalization analysis to identify putative causal genes/SNPs through gene expression. We updated the Methods and Results sections accordingly and summarized the changes below.

Functional annotation of eQTL, pQTL and enhancers

Methods

“Using GTEx database including a set of 49 tissues, we searched for the eQTLs for the genetic variants associated with unclassified HF and its two subtypes at $p < 0.0005$. We obtained protein-quantitative trait loci (pQTLs) from the Fenland study, a genome-proteome-wide association study in 10,708 European-descent individuals. The genome-proteome-wide association study was performed using 10.2 million genetic variants including plasma abundances of 4,775 distinct protein targets measured using the SOMAscan V4 assay in plasma (PMID: 34648354). The SOMAscan assay employs single-stranded oligonucleotides (aptamers) with specific binding affinity to a single protein. We retrieved functional annotations from the Fenland proteo-genomic study for each SNP we identified for unclassified HF, HFpEF, and HFrfEF, matched by chromosomal position and reference allele ($p < 0.0005$). We also searched 193,218 enhancers regions from 295 cell/tissue types from EnhancerAtlas (PMID: 27515742) for all identified sentinel SNPs.”

Results

“For unclassified HF, sentinel SNPs rs6795366 and rs34432450 were not found in the database. We used proxy variants passed GWS threshold with strong LD were for the search. All sentinel SNPs except rs2634073, rs4977575 and rs79329549 showed evidence of eQTLs in at least one issue type. For HFrfEF, all but sentinel SNPs rs2261792, rs56286049 and rs4977575 had significant eQTLs. Identified eQTLs and their tissue types were summarized in Supplementary Table 14. Using the Fenland database, we identified 416 pQTLs ($p < 0.0005$) for identified GWAS loci (Supplementary Table 15). We identified 17, 10 and 1 GWS loci overlapping with human enhancers for unclassified HF, HFrfEF and HFpEF, respectively (Supplementary Table 16).”

TWAS and colocalization

Methods

Genetically Predicted Gene Expression Analysis

Genetically predicted gene expression was estimated using S-PrediXcan (PMID: 29739930), an approach that imputes genetically predicted gene expression (GPGE) in a given tissue and tests predicted expression for association with a trait using GWAS summary statistics. For this analysis, input included results for common variants in our heart failure GWAS and gene-expression references for 48 tissues from GTEx (PMID: 25954001). Our analyses incorporated covariance matrices based on the 1000 Genomes Project European populations to account for LD structure (PMID: 26432245). Bonferroni-corrected significance threshold was 1.93×10^{-7} for these analyses.

Colocalization analysis

The hypothesis that a single variant underlies GWAS and expression quantitative trait loci (eQTL) associations at a given locus (i.e. colocalization) was tested using coloc (PMID: 24830394), a gene-level Bayesian test that evaluates GWAS and eQTL association summary statistics at each SNP at the locus and provides gene- and SNP-level posterior probabilities for colocalization. For this analysis, input included results for common variants in our GWAS and eQTL summary statistics corresponding to the gene-expression references used in S-PrediXcan analysis.”

Results

“Common variants from the different HF subtype GWAS were used to evaluate the association of genetically predicted gene expression levels with HFrEF and HFpEF across 48 tissues using S-PrediXcan. We identified 49 statistically significant ($P < 5 \times 10^{-7}$) gene-tissue combination pairs genetically predictive of HFrEF risk (Supplementary Table 17), including several gene expression levels in HFrEF-related tissues such as *CLCNKA* expression in the coronary artery (5.26×10^{-11}), *PPP1R1B* (3.52×10^{-8}) and *PGAP3* (1.63×10^{-7}) expression in left atrial appendage, *PROM1* (5.57×10^{-8}), *BPTF* (9.70×10^{-8}), and *PGAP3* (1.44×10^{-7}) expression in the left ventricle. Hypergeometric enrichment analysis showed that most enriched gene expression signals (false discovery rate < 0.05) were in three brain tissues, cortex, cervical spinal cord, and substantia nigra. However, we didn’t identify any genetically predicted gene expression levels associated with HFpEF.

Additionally, we used COLOC to identify the subset of significant genes where there was a high posterior probability that the set of model SNPs in the S-PrediXcan analysis for each gene were both causal for gene expression and HF subtypes. This analysis refined our S-PrediXcan analysis by excluding results that may be the consequence of LD between causal SNPs for gene expression and HF subtypes. All six aforementioned gene-tissue pairs significantly associated with HFrEF has high posterior probability ($P_4 > 0.9$) of colocalization, covering five distinct genes expression in coronary artery, left atrial appendage and left ventricle.”

5. Disease-prevalent tissues/cells should be detected by using S-LDSC or DEPICT.

We performed DEPICT analysis and revised the Methods and Results sections as below.

Methods

“Gene-set and pathway enrichment analysis

Gene-set and pathway enrichment analysis was performed using DEPICT for HFrEF and HFpEF with both genome-wide significant SNPs ($p < 5 \times 10^{-8}$) and suggestive signals using a less stringent threshold ($p < 10^{-4}$) (PMID: 25597830). Common SNPs with MAF > 0.01 , HWE $p > 10^{-20}$ and imputation $R^2 > 0.5$ were included in the analysis.”

Results

“To identify pathways and tissues overrepresented in the GWAS of HFrEF and HFpEF, we used the DEPICT gene-set enrichment tool, using all SNPs with p-value less than 10^{-4} for the respective subtype. We identified 4 gene sets significantly associated (false discovery rate <0.05) with HFrEF (Supplementary Table 18) including protein-protein interaction subnetworks. No gene sets were significantly associated with HFpEF using the same approach. We also identified six and six tissue types suggestively associated (false discovery rate <0.2) with HFrEF and HFpEF, respectively (Supplementary Table 19). The top enriched tissue types including heart and endocrine glands for HFrEF, and blood vessels, epithelial cells, and blood for HFpEF.”

6. Line 111-112.

The authors mentioned the difference in the effect direction of sentinel SNPs between HFpEF and HFrEF, What statistical model was used to test this? To show the difference, the Heterogeneity P-value obtained from a fixed-effect meta-analysis could be useful. I also propose that a figure could be intuitive where the value of beta in HFpEF and HFrEF are plotted on the X and Y axes, respectively, and where the dissociation from $Y = X$ is shown.

We tested the associations between the sentinel SNPs and HF subtypes by modeling HFrEF/HFpEF as a binary outcome. The significant association indicates different genetic effects between the two subtypes. As the reviewer suggested, we have generated the figure (see below) to illustrate potential difference in effect size between HFrEF and HFpEF, with standard errors. The blue diagonal line indicates perfect matching in effect size ($X=Y$). Since most loci were only significant in the HFrEF GWAS, we observed that their effect sizes were mostly larger for HFrEF than that for HFpEF except for the *FTO* locus. We also considered the heterogeneity p-value from a fixed-effect meta-analysis (see the table below), which showed consistent evidence from our original comparison. Only two SNPs in *FTO* on Chromosome 16 showed similar effect sizes between two HF subtypes. However, we think the heterogeneity test from the meta-analysis is biased since both HFrEF and HFpEF GWAS shared the same control group without any history of HF. We added the comparison plot (Supplementary Figure 7) and kept our original test results in the revised manuscript.

Results:

“A scatterplot illustrating the comparison between the effect sizes of the GWS loci for HFrEF and HFpEF is shown in Supplementary Figure 7, with effect sizes with standard errors for HFrEF and HfpEF on X and Y axis, respectively.”

rsID	Position	Closest Gene	Heterogeneity P-value
HFrEF			
rs1763610	1:16335527	HSPB7	1.85E-10
rs2261792	1:61881191	NFIA	2.52E-04
rs12612948	2:11568740	E2F6	6.12E-04
rs56286049	3:69824230	MITF	1.57E-04
rs9349379	6:12903957	PHACTR1	8.40E-05
rs4151702	6:36645988	CDKN1A	2.19E-12
rs10455872	6:161010118	LPA	9.49E-03
rs4977575	9:22124744	CDKN2B-AS	5.11E-03
rs2234962	10:121429633	BAG3	6.46E-16
rs7306330	12:51320290	METTL7A	1.18E-05
rs7188250	16:53834607	FTO	0.90
rs3764351	17:37824339	PNMT	4.40E-03
rs4790908	17:65852907	BPTF	2.26E-02
HFpEF			
rs11642015	16:53802494	FTO	0.42

7. Line 113-115.

BAG3 is a famous pathogenic gene for dilated cardiomyopathy. So it was surprising that *BAG3* missense variant (rs2234962) was associated with lower risk for HFpEF. Could you describe a possible reason for this?

The reviewer is correct that the missense *BAG3* variant (rs2234962)/*BAG3* locus has been associated with dilated cardiomyopathy (DCM). The risk allele T was also associated with higher

risk of HFrEF (Table 3) as anticipated given the clinical relationship between DCM and HFrEF. The association between T allele and lower risk of HFpEF can be attributable to its association with quantitative LVEF. The risk allele of HFrEF/DCM is associated with lower LVEF in non-HF population (PMID: 31554410). Since the HFrEF and HFpEF are classified by LVEF (HFrEF: $\leq 40\%$; HFpEF: $\geq 50\%$), the LVEF-associated *BAG3* variant increased the risk for HFrEF (lower mean LVEF) and decreased the risk for HFpEF (higher mean LVEF) comparing to the controls. We also recognize that this is an association analysis and does not directly infer causality. Future investigations of the molecular and pathophysiological functions of *BAG3* gene may further elaborate its potential causal role in HF subtypes.

8. Line 135. "15 autosomal genes related to cardiomyopathy."
Reference 14 is somewhat outdated. The latest one is available (<https://doi.org/10.1038/s41436-018-0039-z>). According to it, there may be more than 15 genes.

We appreciate the reviewer's suggestion. We added the citation of the more recent publication by Hershberger RE et al (PMID: 29904160, 2018) focusing on cardiomyopathy. This publication cited the ACMG SFv2.0 (PMID: 27854360) for 16 genes related to cardiomyopathy including X-linked gene *GLA* (galactosidase alpha), which is not included in our present GWAS of HF. We added additional genes listed in Hershberger RE et al paper (PMID: 29904160) and revised the Results and Supplementary Table 7.

Results:

"Among 27 autosomal genes related to cardiomyopathy (PMID: 29904160), we found significant associations in HFrEF with *TMEM43*, *BAG3*, *MYBPC3*, *TTN*, and in HFpEF with *DSG2* and *PRKAG2* (Supplementary Table 7, Supplementary Figure 7)."

9. Line 144-151. Associations of HFrEF- and HFpEF Loci with Cardiovascular Risk Factors
I wonder if the summary statistics of UKBB published by Ben Neal's lab would be better to compare without bias caused by study differences. Also, is it possible to look at associations comprehensively without limiting them to cardiovascular risk factors?
Besides, it seems that one variant was determined to see the association. I think colocalization analysis should be performed to determine the association.

UK Biobank is a great genomics resource as a large study with relatively homogeneous phenotypes. However, its predominant British White participants may not fully represent the European ancestry for genetic effects, and its sample size is substantially smaller than recent GWAS meta-analysis of European ancestry. Thus, our summary of genetic associations with HF risk factors is appropriate to represent how the identified HFrEF/HFpEF loci related to known HF risk factors. In addition to this targeted genetic association summary of HF risk factors, we extended the search for genetic associations with ~2,400 traits tested in the UK Biobank using the PheWeb browser (<https://pheweb.org/>). All associations with $p < 1 \times 10^{-6}$ were listed in the Supplementary Table 9 to complement Figure 4 which focuses on HF risk factors. We have revised the results section as below.

Results:

“Genome-wide significant loci for unclassified HF and subtypes associated with ~2,400 traits tested in the UK Biobank (searched in PheWeb browser, <https://pheweb.org/>) with $p < 1 \times 10^{-6}$ are listed in the Supplementary Table 9.”

We performed colocalization analysis and other fine-mapping analyses to understand potential causal effects within identified loci. Please see more details in the response to comment #4.

10. Lines 159-168. Mendelian Randomization Association Analysis of HF Risk Factors

Like 8, I wonder if it would be better to select instrumental variables from summary statistics of the UKBB Ben Neal lab provided, instead of taking it from a previously published GWAS study with a different background, so that the background of the study does not become a bias and we can explore more traits. I also suppose it would be interesting to consider both cases “outcome is HF” and “Exposure is HF.”

UKBB is a great genomics resource with relatively large sample size and relatively homogeneous phenotypes, but not without bias and limitations. Its predominant British White participants may not fully represent the European ancestry for genetic effects, and its sample size is substantially smaller than recent GWAS meta-analysis of European ancestry, particularly for most RFs investigated in the present study. Since our current hypothesis-testing design of the MR analyses focused on known HF risk factors in relation to HF subtypes, we are limited to identify novel HF risk factors and won't benefit from the breath of the UK Biobank GWAS summary provided by Neal's lab. Thus, we think the selection of genetic instrumental variables from recent GWAS of European ancestry is appropriate for testing our research hypotheses. We agree with the reviewer that bi-directional MR would be very interesting to investigate. However, the present study is limited to provide conclusive results for “exposure is HF.” Current GWAS results are better powered for testing potential causal effects of HF risk factors on HFrEF and HFpEF. Our GWAS of HFrEF and HFpEF were based on a relatively smaller number of cases than other common diseases and risk factors, and only identified 13 and 1 GWS loci, including the highly pleiotropic *FTO* locus, as potential genetic instrumental variables. We think a future bi-directional MR with more and stronger genetic instrumental variables for HFrEF and HFpEF would produce more robust evidence for potential causal effects of HF subtypes. Therefore, we decided to focus on more robust one-directional MR between risk factors and HF and subtypes.

11. Lines 173-175, 183-185.

The authors speculated that the reason why there were fewer significant loci in HFpEF compared to HFrEF was due to the Heterogeneity of HFpEF. And, as an example, they cited a stroke GWAS. However, I have not been convinced by this example. Please explain in more detail the rationale for assuming that heterogeneity within the disease, like stroke, is the cause of the fewer loci in HFpEF. It may simply be that the heritability is small due to large environmental factors.

The reviewer raises a very important point that we recognize needs further clarification. Our results showed similar heritability between HFrEF (h^2 of 0.25-0.34) and HFpEF (h^2 of 0.22-0.29) in comparable liability scale (see more details in response to minor comment #5). Such results of

heritability estimate cannot fully explain drastically different numbers of GWS loci from HFrEF and HFpEF GWAS. Our conclusions are based on the results of the association analyses when we separated the HF cases into the more refined sub-categories of HFrEF and HFpEF. The limited genetic discovery in unclassified HF despite a large uniformly phenotyped cohort was similar to that seen in the other GWAS of a large cohort. One of the reasons is the intermixing of HFpEF and HFrEF in unclassified HF. On resolving this heterogeneity and repeating GWAS of HF subtypes, we found increased genetic discovery in HFrEF clearly suggesting the importance of accurate and homogeneous phenotype in facilitating genotypic discovery. However, in spite of a large and well-curated sample of HFpEF (about the same sample size as HFrEF) to conform to current clinical definition of HFpEF (PMID: 3537950), we found only one associated locus which was related to BMI. We acknowledge that there may be several reasons for this finding, considering the complicated molecular and pathophysiological mechanisms between genotype and clinical phenotype. We based our statement on current literature that also suggests that HFpEF is a heterogeneous disease phenotype. (PMID: 35767073, 35679364) Multiple clinical trials of pharmaceutical agents have not led to specific treatments for HFpEF, for which heterogeneity has been proposed as the most likely explanation. (PMID: 31926856) We have clarified these issues in the revised manuscript indicating that this is the most plausible explanation, while other issues such as the high prevalence of comorbidities could also play a role in our findings, affecting the associations more in HFpEF than HFrEF.

Discussion:

“There is a developing consensus that HFpEF as currently defined may not represent a cohesive pathophysiology, rather, that HFpEF represents a heterogeneous entity comprised of multiple phenotypes. (PMID: 35767073, 35679364) Multiple large randomized clinical trials utilizing medications that were found to be effective in pre-clinical models of HFpEF did not demonstrate beneficial effects. (PMID 31475794) The contrast with HFrEF is a major reason to conclude that HFpEF may be a heterogeneous disease. Although both HFpEF and HFrEF are associated with risk factors and comorbidities, animal models of HFrEF have identified drug targets which have been conclusively proven to reduce morbidity and mortality by large clinical trials.(PMID 35379504) This is in stark contrast to HFpEF, in which the animal models while recapitulating the cardiac pathophysiology, have failed to identify drug targets that benefit HFpEF, suggesting that the pathophysiology of HFpEF may not be as uniform as seen in HFrEF. Our study also showed that despite increased phenotypic refinement of unclassified HF into HFpEF and HFrEF cohorts of similar size, the yield of GWS loci in HFpEF was even lower than in unclassified HF and in contrast to the increased genetic discovery in the HFrEF cohort. We recognize that this does not directly translate into a conclusion of pathophysiologic heterogeneity since many factors influence the pathway from genotype to phenotype, and it is also possible that appropriate drug targets have yet to be identified for HFpEF; however these findings do suggest that pathophysiologic heterogeneity may have played a significant role in our findings. Our findings suggest an urgent need to develop consensus sub-phenotyping strategies to resolve the heterogeneity of HFpEF as currently defined, as will be the focus of the recently initiated National Institutes of Health HeartShare Program (<https://grants.nih.gov/grants/guide/rfa-files/RFA-HL-21-015.html>).”

12. The authors utilized natural language processing to exclude HF with improved EF. I think the validation of natural language processing is required (eg. compare results obtained in and outside

VA).

We have included more description of our validated phenotyping methods in the revised manuscript to address this comment. NLP was utilized to extract EF values from echocardiogram reports, imaging reports, and clinical notes to ensure that we captured all recorded ejection fractions. The ejection fraction capture algorithm was validated independently of HF diagnosis (PMID: 28606104) to ensure EFs were captured accurately. This method to ensure accurate capture of HFpEF to the exclusion of HF with recovered EF was further validated in the previously reported work (PMID: 29899018). We have clarified these issues in the revised manuscript.

Methods:

“HF patients were identified as those with an International Classification of Diseases (ICD)-9 code of 428.x or ICD-10 code of I50.x and an echocardiogram performed within 6 months of diagnosis (median time period from diagnosis to echocardiography was 3 days, interquartile range 0-32 days). Since the accurate classification of HF into HFrEF and HFpEF is dependent on capture of LVEF values, we used a comprehensive approach based on natural language processing (NLP). As previously described, an NLP tool was developed and validated in the national VA database to extract LVEF values from echocardiogram reports (PMID: 28606104). We utilized NLP to capture LVEF values from nuclear medicine reports, cardiac catheterization reports, history and physical examination notes, progress notes, discharge summary notes, and other cardiology notes, to ensure that we captured LVEF values measured outside the VA. (PMID: 29899018) Using analysis of patient records by blinded physician reviewers, we validated the accuracy of the NLP algorithms to capture LVEF and correctly classify HFpEF. (PMID: 29899018). Compared to our previous studies, we utilized a wider time frame between HF diagnosis and first recorded LVEF for this study to ensure that we captured LVEFs recorded outside the VA soon after HF diagnosis but entered into the VA medical records later. We classified HFpEF as presence of HF diagnostic code and first recorded EF of $\geq 50\%$ and HFrEF as HF diagnostic code with first recorded LVEF of $\leq 40\%$.”

Minor comments

1. The number of cases differed in figure 2.

Line 81. HFpEF case = 19,589.

In figure 1, HFpEF case = 19,589.

In figure 2, HFpEF case = 19,598.

We corrected the typo in Figure 2 as 19,589.

2. Line 97. "established definitions of HFrEF and HFpEF"

The definition of HFpEF and HFrEF using the EF threshold is often used and prevalent, but I do not think “established”.

We agree with the reviewer and have revised the manuscript to indicate that the EF thresholds used are current consensus approaches and not established.

“We conducted GWAS in cohorts of HFrEF and HFpEF curated based on the current LVEF criteria.”

3. Line 114, 132.

Is this just because HFpEF and HFrEF are competing risks to each other?

line 113-116: “*BAG3* missense variant (rs2234962) was associated with higher risk for HFrEF (OR 1.12, 95% CI 1.09-1.15, p-value 9.02×10^{-18}), but was associated with lower risk for HFpEF (OR 0.97, 95% CI 0.94-0.99, p-value 6.42×10^{-3}).”

line 132-134: “Interestingly, the sentinel SNP of the *FTO* locus was significantly associated with HFpEF (rs11642015, OR 1.10, 95% CI 1.03-1.17, p-value 6.30×10^{-3}), but not associated with HFrEF (rs7188250, OR 1.06, 95% CI 0.99-1.12, p-value 0.11).”

We thank the reviewer for pointing out these interesting observations. However, we don’t have evidence to support the explanation of competing risk. In the case of *BAG3* locus, the risk allele of HFrEF is associated with lower LVEF in non-HF population (PMID: 31554410). Since the HFrEF and HFpEF are classified by LVEF (HFrEF: $\leq 40\%$; HFpEF: $\geq 50\%$), the LVEF-associated *BAG3* variant increased the risk for HFrEF (lower mean LVEF) and decreased the risk for HFpEF (higher mean LVEF) comparing to the controls. For the *FTO* locus, the difference between the associations with HFrEF and HFpEF in African Americans is more likely due to different local LD structure between the European vs. African ancestries since the genetic associations with HFrEF and HFpEF were similar among European Americans (Table 3).

4. Line 120-121.

It would be easier to understand if there is also a “ratio” value obtained from LDSC to show the inflation-polygenicity ratio.

We thank the reviewer for the suggestion. The ‘ratio’ value obtained from LDSC measures the proportion of the inflation in the mean χ^2 that the LD Score regression intercept ascribes to causes other than polygenic heritability. A small ratio is anticipated in the GWAS, though in practice values of 10-20% are not uncommon, especially when the overall inflation is moderate. We added the values of the inflation-polygenicity ratio in the revised manuscript.

“The LDSC ratios for unclassified HF, HFrEF and HFpEF are 0.1381 (SE of 0.0295), 0.0723 (SE of 0.0456) and 0.2184 (SE of 0.0592), respectively.”

5. Line.153-157. Heritability & Genetic Correlation

Calculation with LDSC is based on summary statistics. Since raw data is available, would there be any difference if it is calculated with GREML?

LDSC-based calculation typically underestimates the heritability. We have added the heritability estimates using GREML which showed substantial higher heritability than the LDSC-results. We presented heritability estimates based on GREML in the revised manuscript.

Methods

“We used GREML-LDMS-I as implemented in Genome-wide Complex Trait Analysis (GCTA) 1.93.0beta to estimate the multicomponent heritability of unclassified HF, HFrEF and HFpEF in our MVP participants of European ancestry. GREML-LDMS-I was shown to be the least biased and one of the most accurate heritability estimation methods (PMID: 29700474). Restricted by computing memory requirements, we randomly selected 50,000 unrelated MVP non-Hispanic Whites to perform GREML-LDMS-I analysis (PMID: 22344220, 21376301). We then estimated heritability within each group after applying identical quality control procedures. SNPs that were multi-allelic, had MAC < 6, or call-rate < 95% were removed. LD scores were computed on each autosome using an r^2 cutoff of 0.01, and the genome-wide LD score distribution was used to assign SNPs to 1 of 4 LD quartile groups, where groups 1-4 represented SNPs with higher LD scores. Within each LD group, SNPs were further stratified into 6 MAF bins ([0.001, 0.01], [0.01, 0.1], [0.1, 0.2], [0.2, 0.3], [0.3, 0.4], [0.4, 0.5]) and a genetic relatedness matrix (GRM) was constructed from each bin, creating 24 GRMs. Finally, GCTA --reml was used to fit a model of HF case status based on the 24 GRMs, with age and sex as covariates. Total observed heritability estimates were transformed to estimate disease liability scale across a range of presumed HF subtypes prevalence estimates (2.5% to 7% for each HF subtype).”

Results

“We estimated the SNP-based heritability using GREML-LDMS-I in MVP non-Hispanic Whites. Assuming a prevalence of HFrEF and HFpEF of 2.5%, 5%, and 7.0% in the population, we derived similar heritability on the liability scale between HFrEF (0.25, 0.31, 0.34, respectively) and HFpEF (0.22, 0.26, 0.29, respectively) (Supplementary Figure 9).”

6. Lines 210-213. "Hence our findings indicate that the issue with reduced genetic discovery in our cohort was not secondary to the impurity of the phenotype due to EMR-based curation, but rather that HFpEF as currently defined may be a collection of sub-phenotypes with multiple independent disease mechanisms."

It would be appreciated if there are proposals on how to do this as well as the need for subclassification for HFpEF. Any such hints from clinical studies? What about from the genome side?

We thank the reviewer for this comment. As the reviewer alludes to and we have elaborated in response to major comment #11, most of the evidence for heterogeneity is derived from the inability of large clinical trials of treatment strategies, which had neutral results, suggesting that the therapies may have had positive results had they been targeted to specific sub-groups within HFpEF. However, as the reviewer suggests, this needs stronger evidence. A priori sub-phenotyping based on clinical characteristics have been utilized in HFpEF – such as HFpEF with pulmonary hypertension or with CKD as separate sub-phenotypes. But as the reviewer indicates, considering the large number of co-morbidities that associate with HFpEF, artificial intelligence approaches might be the best method to resolve the heterogeneity, and a few small studies have demonstrated the potential of this approach. We are not aware of any other large genomic analyses of HFpEF; however, we agree with the reviewer that incorporating -omics in machine learning approaches to complement clinical data would be crucial to resolve the heterogeneity of HFpEF and establish possible pathophysiology underlying such sub-phenotypes.

Discussion

“Initial studies that applied unsupervised clustering approaches to clinical and biomarker data mainly derived from HFpEF clinical trials suggest that different sub-phenotypes may underlie HFpEF.(15-18) For example, Cohen and colleagues used latent class analysis on data from the

TOPCAT Trial (Treatment of Preserved Cardiac Function Heart Failure with an Aldosterone Antagonist Trial) and identified three sub-phenotypes of HFpEF, with one of the sub-phenotypes associated with better response to spironolactone.(16) Based on these initial results, it is possible that artificial intelligence/machine learning approaches applied to clinical, imaging, biomarker and -omics data may identify specific sub-phenotypes of HFpEF that may be benefited by specific drug therapy. While artificial intelligence / machine learning approaches applied to clinical and biomarker data may resolve some of the heterogeneity of HFpEF, biologically based approaches to address the potential for rare genetic variants to influence disease pathogenesis and the complexity of the path from genotype to phenotype using multi-omics, epigenomics and chromatin dynamics, and single cell approaches, may be needed to truly uncover the pathobiology of HFpEF.”

7. Line 212.

EMR (electronic medical records?) should be spelled out because it was not defined anywhere in this manuscript.

We apologize for this oversight and have used EHR throughout the manuscript.

8. Line 247. "measurement of B-type natriuretic peptide at baseline"

It is not essential to measure BNP to diagnose HF. A high BNP value should be a clue. So, please explain why just measuring BNP can be a criterion to diagnose HF.

The reviewer is correct that we did not use BNP level to establish diagnosis. We have clarified this detail in the Methods section.

Methods

“In the algorithm for identification of HF patients, we used documentation in EHR of the ordering of B-type natriuretic peptide as one of the criteria, since evaluation of practice patterns indicated that ordering of B-type natriuretic peptide increased the likelihood of the patient having clinical HF, as validated by blinded review. (PMID: 29899018).”

9. Line. 264-266. " We excluded: duplicate samples, samples with more heterozygosity than expected, an excess (>2.5%) of missing genotype calls, or discordance between genetically inferred sex and phenotypic gender."

Since the association tests were performed based on a logistic regression model implemented in PLINK2, related individuals in addition to duplicated ones should be excluded?

We thank the reviewer for pointing this out. We did exclude related individuals in the present study similar to previous MVP studies (e.g., PMID: 30275531). Right after the quoted sentence, we stated that “In addition, one individual from each pair of related individuals (more than second degree relatedness as measured by the KING software) were removed.” We would be happy to include published technical details if needed.

10. Whilst the main theme of this study is the genetic difference between HFpEF and HFrEF, I felt that the consideration and comparison with African Americans were insufficient.

As the reviewer mentions, the main theme of the study was to examine genetic differences between HFrEF and HFpEF. As the reviewer points out, examining genetic bases of HF in African Americans is very important and will be the focus of future studies as we have more enrollment of African Americans and more genomic data with improved imputation coverage. We performed preliminary GWAS investigations and identified two GWS loci of HFrEF located on chromosome 6 (*CDKN1A*) and 11 (*DDB2*). No GWS loci were identified for composite HF or HFpEF. We will collaborate with other independent cohorts to conduct a multi-ancestry GWAS of heart failure subtypes including additional samples of African ancestry to improve power of ancestry-specific and cross-ancestry discovery.

11. Figure 1 and Figure 2, and Tables S4 and S5 are in the opposite order compared to the main manuscript.

We revised the manuscript and Figures/Tables to make them in the correct order.

Reviewer #2 (Remarks to the Author):

1. The authors describe a GWAS study of heart failure (HF) comparing reduced (HFrEF) versus preserved (HFpEF) ejection fraction. This is an interesting study. A genetic comparison of these two clinical subtypes of heart failure is an important and not insignificant undertaking.

It is strongly argued the pathological heterogeneity of HFpEF explains the largely failed outcomes of clinical trials. The authors believe clinically defined HFpEF represents an amalgamation of several different pathobiological entities and highlight the need to sub-phenotype it. Even with machine learning, it is not clear how this might be easily accomplished, and there could be multiple reasons for failed clinical trials, including lack of suitable targets!

We agree with the reviewer that we cannot firmly conclude that proper understanding of the pathophysiology and management of HFpEF is impeded by heterogeneity. We have revised the manuscript in response to these comments to indicate that heterogeneity is a very likely possibility and not completely proven by current studies as detailed in the response to Major comment #11 and Minor comment #6 from Reviewer 1.

2. This is an impressive GWAS, but have quantitative GWAS approaches been considered? The samples provided by the Million Veteran Program appear to have been well phenotyped, but the description of phenotyping is vague. Notwithstanding this, the authors replicated all 12 SNPs associated with HF, previously published in Nat Commun (Shah et al., 2020). The associations with unclassified HF identified in patients of European ancestry were then replicated in African Americans. The authors follow up with a Mendelian randomization study linking well-known risk factors with HFrEF and HFpEF.

We have clarified the phenotyping methodology in the revised manuscript as detailed in response to major comment #12 and Minor comments # 2 and 8 from Reviewer 1.

3. The authors casually mention using the less restrictive definitions for HFpEF because it is better powered, but this needs a stronger justification. The authors could also have discussed the heritability (h^2) scores.

We apologize for the lack of clarity of the phenotyping approaches and have described these in more detail. In previous observational studies, we used specific additional criteria to define HFpEF which led to a significant reduction in the number of patients with HFpEF. For the current study we used the clinical criteria (HF code + EF during diagnosis $\geq 50\%$) and compared the cohort with these criteria to the cohort that was curated using the more restrictive criteria ($n=12,119$ cases) in terms of genetic associations. We found that genetic associations were similar but higher p-values in restrictive HFpEF definition (see figure below) indicating that the population genomic structure was similar and hence we used the larger cohort for our analyses. We added a section in the Discussion to justify our HFpEF definition in the GWAS. We included additional results and discussion about heritability of HF subtypes using GREML method as described above.

Discussion

“In addition, to ensure that our findings were not due to issues in curating the HFpEF phenotype from the EHR, we used the more restrictive phenotype utilized in our previous epidemiologic studies based on measurement of natriuretic peptides and use of diuretics which had a positive predictive value of 96% on blinded analysis¹² and repeated GWAS in this more restrictively curated sub-group of HFpEF, and found similar genetic associations but less statistical power (due to smaller sample size) comparing to the main HFpEF cohort (Supplementary Figure 10). Using LDSC and the GWAS summary statistics, we found that the genetic correlation between

the two HFpEF definitions was very high ($r=0.981$, $p<2 \times 10^{-16}$). Among top 110 HFpEF-associated common SNPs ($p<10^{-6}$, $MAF>1\%$), the genetic effects between the two HFpEF GWAS were highly correlated ($r=0.995$, $p<2 \times 10^{-16}$). Mostly driven by a larger number of HFpEF cases in the original definition (19,598 vs. 12,119), the p-values of 109 out of 110 SNPs were lower in the original HFpEF GWAS conducted in the less restrictive cohort”

4. Overall, this is a good piece of work, but the potential biological relevance of the loci identified in relation to this and related studies is hardly mentioned. Apart from highlighting the sample size, the authors don't do themselves any justice, by failing to highlight the novelty of this work, and its implications. Perhaps, pathway analysis might have been informative.

We agree with the reviewer that we had not emphasized the biological significance of our genetic discovery in the manuscript and have substantially revised the manuscript to add the biologic relevance of identified loci of HF and subtypes (e.g., eQTL, pQTL, TWAS, colocalization analysis, and DEPICT pathway analysis). The Discussion has also been revised to address this comment as detailed below:

“The novel genetic associations with HFrEF confirm known pathophysiology and also indicate novel biology that merits further investigation. Myocardial remodeling is driven by inappropriate activation of various neurohormonal systems, including the sympathetic nervous system and its effector hormones, the catecholamines epinephrine and norepinephrine. (PMID: 35379504, 28836616) Blockade of the adrenergic beta receptor to decrease action of these hormones has substantially improved survival in HFrEF. The *PNMT* gene encodes phenylethanolamine N-methyltransferase, which catalyzes the N-methylation of norepinephrine to epinephrine. Sequencing of the *PNMT* gene has found several SNPs including non-synonymous SNPs in the coding region that affected transcription (16277617). Previous studies have associated polymorphisms of the *PNMT* gene to catecholamine levels and hypertension. Cui and colleagues found that the allelic frequency of an SNP was different between hypertensives and normotensives among African Americans but not among other ethnic groups. (PMID 14553966), while Huang et al found an association of the risk of hypertension with *PNMT* polymorphisms in Han Chinese population (21866188). Polymorphisms of the *PNMT* gene also influence the levels of post-exercise surge in catecholamine levels (18349382). Our data is the first, to our knowledge, that demonstrates an association of *PNMT* genetic variation with the risk of HFrEF. The gene *E2F6* codes for a member of the E2F family of transcription factors that regulate cardiac development, cardiomyocyte growth, and myocardial metabolism (28964969, 22403008, 28085920). Overexpression of *E2F6* in the mouse myocardium leads to cardiomyopathy (22403008), which is associated with decreased glycolytic activity and increased expression of β -hydroxybutyrate dehydrogenase, an enzyme that regulates ketone metabolism (28085920). In contrast to the deleterious effects of *E2F6* overexpression during cardiac development, *in vitro* studies have shown that *E2F6* may protect against cardiotoxic agents (28964969). Preclinical studies have shown that microphthalmia transcription factor (*MITF*) regulates the hypertrophic response of the myocardium (PMID: 16998588), and that the effect of *MITF* on the myocardial hypertrophic pathway may be mediated epigenetically by the microRNA miR-541 (24722296). Another potential mechanism of action of *MITF* on myocardial hypertrophy is via an interaction with four-and-a-half LIM domain protein (*FLH2*) thereby influencing the expression of ErbB2 interacting protein (*Erbin*) (26025865). While GWAS have shown an association of the

PHACTR1 locus with multiple vascular diseases such as hypertension (30578418) and coronary calcification (22144573), down-regulation of PHACTR-1 function in vascular cells did not lead to vascular pathology in pre-clinical studies (35387477). The transcription factor NFIA, which has major roles in glial cell development, has been associated with ventricular electrical activity (QRS duration on electrocardiogram) by two population genomic studies (PMID: 27577874, 23463857). In a genetic study of renal and cardiometabolic disease in Zuni Indians, NFIA was associated with diastolic blood pressure (25688259). While the function of Methyltransferase Like 7A (*METTL7A*) is not well understood, other methyltransferases such as METTL3 and -14 methylate N6-adenosine moieties in RNA and oppose the action of FTO, a N6-adenosine demethylase, which is the only gene that was significantly associated with HF, HFpEF, and HFpEF (PMID: 28985428); myocardial changes in N6-adenosine methylation of mRNA is associated with progression to HF. (PMID: 31849158).”

5. Clearly, like all other complex diseases, HF is clinically and genetically heterogeneous. However, the focus for genetic studies of HF (and other complex diseases) is shifting, with increasing attention on rare variant analysis, chromatin dynamics, and single cell approaches, which are more likely to deliver new insights than bigger GWASs.

We agree with the reviewer that more detailed analysis of the pathway from genotype to phenotype using whole genome sequencing, analysis of chromatin states, and single cell omics approaches may uncover novel biology underlying HF and its subphenotypes; we have expanded the discussion to include these important points as described in response to Comment #1 and Reviewer 1 Minor comment #6.

REVIEWERS' COMMENTS

Reviewer #1 (Remarks to the Author):

The manuscript has been substantially improved and my concerns have been resolved. Just one thing, data availability (e.g. GWAS summary statistics) should be clearly described in the manuscript.
Anyway, congratulations on the authors' great results for heart failure.

Reviewer #2 (Remarks to the Author):

The authors have substantially addressed my comments and the revised manuscript is much improved. Overall, I hope they found these suggestions helpful and I wish them good luck for their publication.

We thank the reviewers for their comments.

Reviewer #1 (Remarks to the Author):

The manuscript has been substantially improved and my concerns have been resolved. Just one thing, data availability (e.g. GWAS summary statistics) should be clearly described in the manuscript.

Anyway, congratulations on the authors' great results for heart failure.

Response: We thank the reviewer for these comments. We have added the following sentence to the Data Availability statement in the manuscript:

“The full summary level association data genome-wide association analyses in the MVP and the meta-analysis from this report will be available through dbGaP (accession number phs001672).

Link: https://www.ncbi.nlm.nih.gov/projects/gap/cgi-bin/study.cgi?study_id=phs001672.v8.p1”

Reviewer #2 (Remarks to the Author):

The authors have substantially addressed my comments and the revised manuscript is much improved. Overall, I hope they found these suggestions helpful and I wish them good luck for their publication.

Response: We thank the reviewer for these comments.